Corrected: Author correction

# Indirect tail states formation by thermal-induced polar fluctuations in halide perovskites

Bo Wu [1,2], Haifeng Yuan [3], Qiang Xu [2], Julian A. Steele[4], David Giovanni[2], Pascal Puech[5], Jianhui Fu[2], Yan Fong Ng[6,7], Nur Fadilah Jamaludin[6,7], Ankur Solanki[2], Subodh Mhaisalkar [6,7], Nripan Mathews[6,7], Maarten B.J. Roeffaers[4], Michael Grätzel[7,8], Johan Hofkens [3] & Tze Chien Sum [2]

Halide perovskites possess enormous potential for various optoelectronic applications. Presently, a clear understanding of the interplay between the lattice and electronic effects is still elusive. Specifically, the weakly absorbing tail states and dual emission from perovskites are not satisfactorily described by existing theories based on the Urbach tail and reabsorption effect. Herein, through temperature-dependent and time-resolved spectroscopy on metal halide perovskite single crystals with organic or inorganic A-site cations, we confirm the existence of indirect tail states below the direct transition edge to arise from a dynamical Rashba splitting effect, caused by the $PbBr_6$ octahedral thermal polar distortions at elevated temperatures. This dynamic effect is distinct from the static Rashba splitting effect, caused by non-spherical A-site cations or surface induced lattice distortions. Our findings shed fresh perspectives on the electronic-lattice relations paramount for the design and optimization of emergent perovskites, revealing broad implications for light harvesting/photo-detection and light emission/lasing applications.

[1] Institute of Electronic Paper Displays, South China Academy of Advanced Optoelectronics, South China Normal University, Guangzhou, Guangdong Province 510006, China. [2] Division of Physics and Applied Physics, School of Physical and Mathematical Sciences, Nanyang Technological University, 21 Nanyang Link, Singapore 637371, Singapore. [3] Department of Chemistry, KU Leuven, Celestijnenlaan 200F, B-3001 Leuven, Belgium. [4] Centre for Surface Chemistry and Catalysis, KU Leuven, Celestijnenlaan 200F, 3001 Leuven, Belgium. [5] CEMES/CNRS, University of Toulouse, 31055 Toulouse, France. [6] School of Materials Science and Engineering, Nanyang Technological University, 50 Nanyang Avenue, Singapore 639798, Singapore. [7] Energy Research Institute @NTU (ERI@N), Research Techno Plaza, X-Frontier Block Level 5, 50 Nanyang Drive, Singapore 637553, Singapore. [8] Laboratory of Photonics and Interfaces, Department of Chemistry and Chemical Engineering, Swiss Federal Institute of Technology, Station 6, 1015 Lausanne, Switzerland. These authors contributed equally: Bo Wu, Haifeng Yuan. Correspondence and requests for materials should be addressed to J.H. (email: johan.hofkens@chem.kuleuven.be) or to T.C.S. (email: Tzechien@ntu.edu.sg)

Solution-processed lead halide perovskites are the rising stars in photovoltaics (PV), light-emitting devices (LEDs), and photodetectors[1]. A clear understanding of their electronic–lattice relations is the key to harnessing the full potential of their amazing optoelectronic properties such as long bipolar carrier diffusion and high charge separation efficiency[2,3]. Several theories such as large polaron formation[4], ferroelectric effect[5,6], organic cation screening[7,8], and Rashba effect[9,10] have recently been proposed to account for these properties. Within the experimental context, spectroscopic characterization remains one of the most important techniques to pinpoint the electronic–lattice relations through probing light–matter interactions. However, significant knowledge gaps exist in fully describing these interactions; namely, the dual emission in many perovskite single crystals (SCs) is frequently reported, but its origins and mechanisms remain controversial[11–18]. Many reports simply attributed it to the reabsorption effect. Although the reabsorption from structural fluctuations induced Urbach tail states is common for conventional polar semiconductors, it does not give rise to distinct dual emissions. Wang et al.[19] recently attributed the dual emission to the Rashba effect arising from the centro-symmetry breaking by the organic cation. However, the origin of the Rashba effect is far from being understood and this conclusion needs to be carefully relooked. Furthermore, there are several striking discrepancies in the photophysics exhibited by perovskite crystals with differing morphologies and sizes: (1) large spread of exciton binding energies[20,21]; (2) contrasting photoluminescence quantum yields[22]; (3) three orders difference in electron–hole recombination rate coefficients ($10^{-11}$ to $10^{-8}$ $cm^3\,s^{-1}$) for different sized and processed perovskite crystals[19,23,24]. Morphological effects have often also been conveniently censured to account for these disparate effects. Strictly, from the Physics viewpoint, the structure–function (or lattice–electronic) relationship in these differently sized and processed perovskites remains vague and confusing.

Herein, we explicate the lattice–electronic properties in a family of perovskite SCs and polycrystalline (PC) film samples with different A cations ($CsPbBr_3$, $FAPbBr_3$, and $MAPbBr_3$) using a broad range of temperature-dependent and time-resolved optical spectroscopies, correlated with density functional theory (DFT) and molecular dynamics (MD) calculations and electrical characterizations. We clarify the dual emission is a general characteristic of all lead halide perovskites, but the relative strength of the low-energy peak is highly dependent on the local environment and may become non-obvious in some cases. We point out that reabsorption based on the conventional Urbach tail concept (i.e., direct transition) and other possible origins are inadequate in universally explaining the dual emission properties. Instead, we establish that the origin should arise from momentum and/or spin forbidden tail state transitions, existing in the presence of a local electric field which can generate a spin-split and momentum mismatch at the conduction band minimum via the Rashba effect. We discerned the origins of the Rashba splitting over different temperature regimes. At low temperatures, non-spherical A-site cations and surface/defects-induced lattice distortion lead to a static centrosymmetry breaking that mainly contributes to the Rashba effect. At high temperatures, when the phonon occupation number increases, the Rashba effect is mainly contributed by thermal $PbBr_6$ octahedra polar fluctuation that breaks the centrosymmetry, regardless of the A-site cation species. These tail states below the direct transition edge with low transition probabilities lower the electron–hole recombination rate—thereby stifling high-efficiency LED operation or optical gain in perovskite SCs. Conversely, the carrier lifetimes are prolonged by the indirect band edge, and the exciton dissociation is aided by the lattice fluctuation, which may be beneficial for PV or photodetectors. Lastly, we conclude the presence of the indirect tail states and the susceptibility of the octahedral deformation to the local environment may lead to the observed large spread of exciton binding energy, photoluminescence quantum yield, and electron-hole recombination coefficient. Our findings rationalize the structure–function relationship of lead halide perovskites, permitting the interpretation of several anomalous optoelectronic properties which have far-reaching implications for their applications and the design of emergent halide perovskites.

## Results

**Universal spectral features of perovskite SCs.** We begin with an overview of the common dual emission nature in lead bromide-based perovskite semiconductors, where Fig. 1a–c shows the temperature-dependent photoluminescence (PL) and diffuse reflectance spectra of $CsPbBr_3$, $FAPbBr_3$, and $MAPbBr_3$ SCs. Their optical images and XRD patterns are given in Supplementary Figure 1. All three SCs show two distinct PL peaks, which we denote as Peak 1 (high energy) and Peak 2 (low energy). We had previously performed diffuse reflectance spectroscopy (DRS) to measure $MAPbI_3$ SC absorption edge and the results were consistent with those of the photo-thermal diffraction spectroscopy reported elsewhere[14,19]. Here DRS is used to measure the absorption properties of the lead bromide perovskites and the absorption edges are consistent with the positions of Peak 2 and its temperature dependence (Supplementary Figure 2). Figure 1d shows the absorption and emission spectra for the representative $MAPbBr_3$ system, which can be strikingly different between SC and PC thin films (TFs—precursor concentration-dependent—see later discussion): two absorption edges (or emission peaks) for SC versus one for TF. Peak 1 corresponds well with the single TF emission peak position. The band edge measured using DRS agrees with the PL position, indicating band edge emission in the TF. Peak 2 is not prominent in TF due to its weak absorbing and emitting nature. The EQE profile of a 0.5 mm $MAPbBr_3$ SC-based photodetector shows high efficiency near Peak 2, indicating weak transition and delocalized nature of the excited species. The absorption is weak, so the light can easily penetrate deep into the crystal to reach the back side. Both carrier types can, therefore, be collected at their respective electrodes. Another requirement for the efficient collection of carriers near Peak 2 is that they should be highly delocalized (free carriers) with long diffusion lengths. Hence, this excludes subgap localized states as a possible origin for Peak 2. Figure 1e shows the temperature-dependent PL peak positions of $CsPbBr_3$, $FAPbBr_3$, and $MAPbBr_3$ SCs versus their TF counterparts extracted with the Voigt function. The anomalous blueshift of Peak 1 with increasing temperature (contrary to conventional semiconductors) is attributed to the stabilization of their out-of-phase band-edge states ($R$ point in the Brillouin zone) with lattice expansion[5]. However, Peak 2 exhibits a redshift with increasing temperature, which can be well-fitted by the Varshni equation for most inorganic semiconductors, accounting for the electron–phonon coupling effect (Supplementary Figure 3). Despite the orthorhombic to cubic phase transitions[25] around 150 K for $MAPbBr_3$ or 160 K for $FAPbBr_3$ (seen as "kinks"—Fig. 1e), the observed temperature-dependent trends for both Peaks 1 and 2 are universal across the family of SCs, therefore discounting phase transitions as the origin. Lastly, both PL peaks in the SCs exhibit a quadratic dependence on excitation power (Fig. 1f), indicating that free electron–hole radiative recombination dominates the emission process.

**Inadequacies of the reabsorption effect.** As mentioned, the origins of the dual PL peaks in perovskite SCs (and also reported

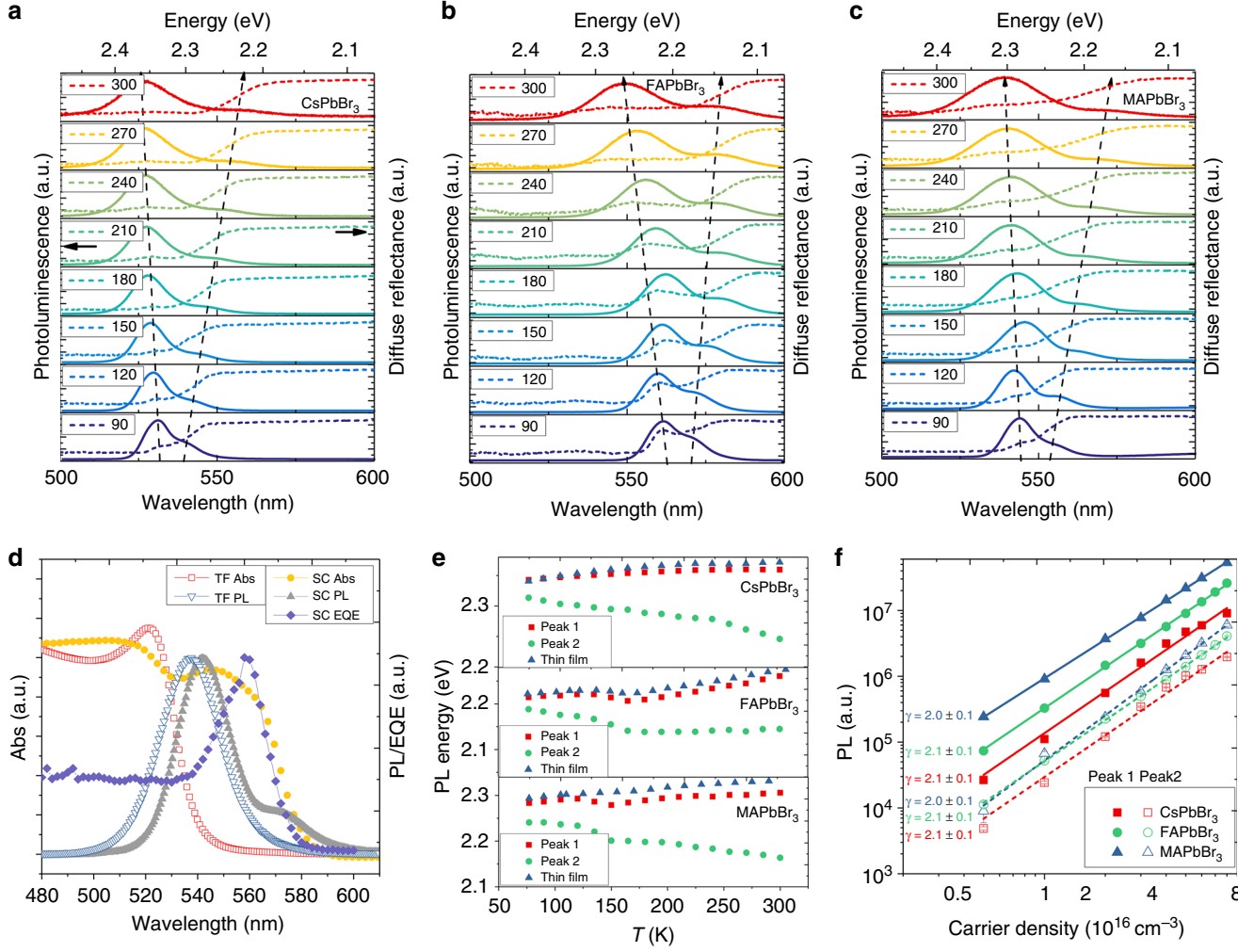

**Fig. 1** Absorption and emission properties of lead bromide single crystals. Temperature-dependent PL and diffuse reflectance spectra of **a** CsPbBr$_3$, **b** FAPbBr$_3$, and **c** MAPbBr$_3$ SCs from 90 K to room temperature (300 K). Dashed arrowed lines are guides to the eye on the evolution of PL Peak 1 (high energy) and Peak 2 (low energy) with temperature. **d** Comparison of the absorption and PL spectra of MAPbBr$_3$ SC and TF at room temperature. The TF was prepared using 0.25 M precursor concentration as discussed in detail in the next section. EQE profile of a photodetector based on a 0.5 mm MAPbBr$_3$ SC is also shown. Solid symbols: SC, open symbols: TF. **e** PL peak positions versus temperature for different lead bromide perovskite SCs and TFs. **f** All the PL peaks in lead bromide perovskite SCs show quadratic dependence on excitation power over the excitation range of 0.5 to 8 × 10$^{16}$ cm$^{-3}$

for PC TF) are still under intense debate. Explanations range from (1) lateral inhomogeneities and coexistence of structural phases[13,18]; (2) defect bound carriers or bound exciton emission[12]; (3) surface versus bulk emission[14,15]; (4) free carrier versus exciton emission[17]; (5) phonon replica[26]; (6) inelastic exciton–exciton scattering (P emission)[27], exciton–electron scattering (H emission) or biexciton emission; and (7) the reabsorption effect[16,28–30]. However, none can satisfactorily account for the dual PL peaks and its universal nature. For example, the temperature-dependent energy shift of Peak 2 cannot be predicted by (1), (3), (4), (5), and (6), and the quadratic intensity dependence of Peak 2 will exclude explanations dependent on (2) and (4). A succinct review of these viewpoints and their inadequacies in light of our latest results are elaborated in Supplementary Note 1. Specifically, the reabsorption effect by tail states (7) appears most compelling and superficially agrees with our observations in Fig. 1. However, such dual emission phenomenon has not been reported in polar semiconductors with typical Urbach tail states. Furthermore, detailed tests on TFs prepared with varied precursor concentration (while maintaining the 1:1 ratio for PbBr$_2$ and methylammonium bromide (MABr)) exclude reabsorption effect as the origin—Peak 2 also emerges in TFs with

increasing precursor concentration (Fig. 2a). The quality of the films improves with increasing precursor concentration as seen in the SEM images (Supplementary Figure 4a), as well as improved carrier lifetimes approaching that of SCs (Supplementary Figure 4b). This implies that Peak 2 may become prominent as the crystalline quality of the perovskite improves. Figure 2b shows that the two TF peaks possess similar temperature-dependent trends as those in SCs (Fig. 1e). Likewise, both peaks also show quadratic power dependence (Supplementary Figure 4c, 4d), due to free electron–hole recombination. Furthermore, the optical scattering related artefacts are unlikely in the TFs as they look very smooth from AFM images with typical root mean square (RMS) roughness less than 8 nm in a 10 μm × 10 μm region (Supplementary Figure 5). These observations hint towards a common intrinsic origin, as seen in the SCs.

Figure 2c shows a schematic illustration of the initial carrier distribution under various excitation and PL collection geometries. Using these different schemes: (1) 400 nm excitation-backscattered collection (400 nm BS), (2) 400 nm excitation-forward-scattered collection (400 nm FS), and (3) 800 nm (two-photon) excitation-backscattered collection (800 nm BS), we show that the PL profiles are similar (Fig. 2d). This indicates

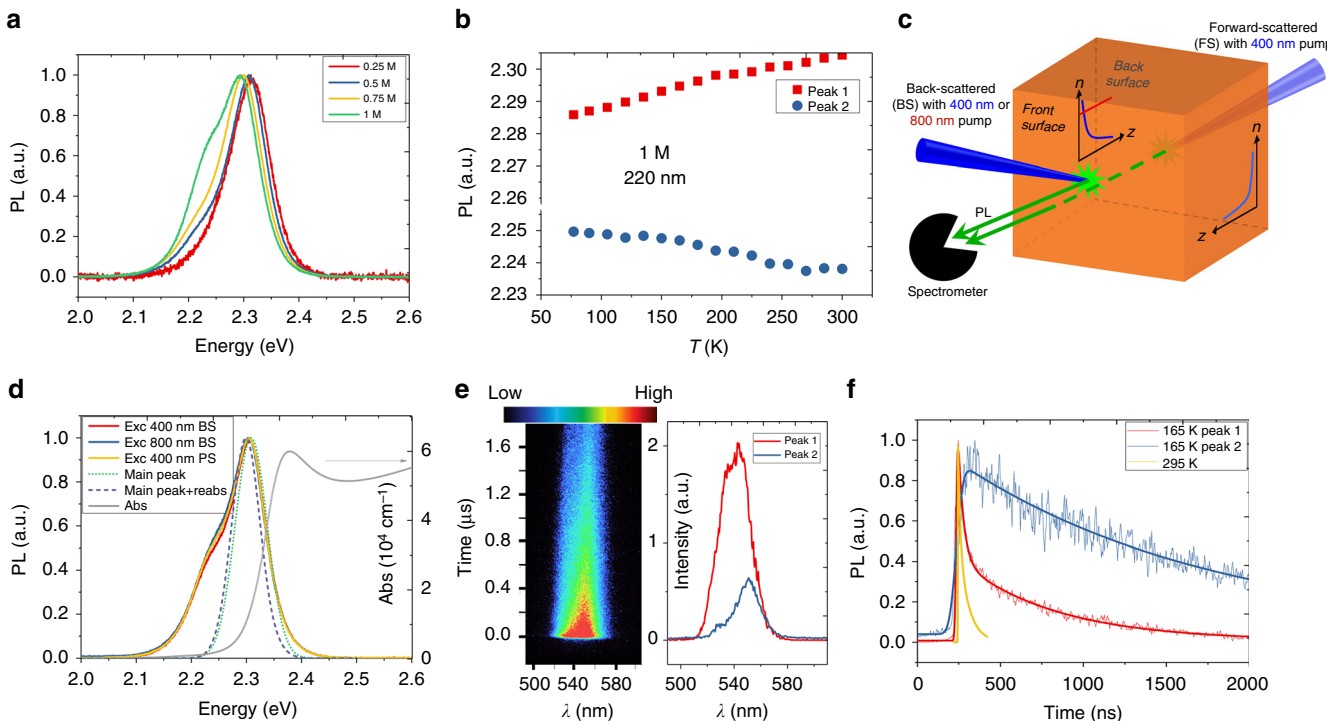

**Fig. 2** The PL properties of the MAPbBr$_3$ polycrystalline (PC) films prepared with different precursor concentrations. **a** PL profiles of the MAPbBr$_3$ PC thin films prepared with 0.25, 0.5, 0.75, and 1 M precursor concentrations. **b** Temperature-dependent PL peak positions for the film prepared with 1 M precursor concentration. This concentration matches that used for single crystal growth. The film has a thickness of around 220 nm. **c** Schematic of different PL excitation and collection geometries for the MAPbBr$_3$ thin films as described in the main text. **d** PL profiles with different excitation and collection schemes as well as fittings of the PL profiles by only considering the reabsorption effect in MAPbBr$_3$ thin films. **e** The pseudo-color mapping of the PL intensity versus time and wavelength at 165 K (left) and the deconvolved peaks using non-negative matrix factorization method (right). **f** The deconvolved PL kinetics at 165 K and the global PL kinetics at room temperature (295 K)

negligible reabsorption effect and/or much faster carrier diffusion and redistribution than the emission rate. The latter is excluded by our calculations based on uniform carrier distribution involving multi-reflection and multi-reabsorption effects[31]. The calculations also reveal that reabsorption could only account for less than 10 meV PL energy shift in this case for a thin 220 nm perovskite film (see Supplementary Note 2 for details), reinforcing our main point that the reabsorption effect is inadequate in solely accounting for the dual peak emission.

**Static versus thermal lattice distortions and indirect tail states.**
Recently, several theoretical works predicted the presence of the Rashba effect in organic–inorganic lead halide perovskites[9,32–35]. The Rashba effect is the consequence of the strong spin–orbit coupling (SOC) and the breaking of inversion symmetry in the crystal in a direction orthogonal to a $k$-point sampling plane, which leads to spin- or momentum-forbidden transitions at the band edge[34]. The Rashba band splitting at the VBM has been directly verified with angle-resolved photo-electron spectroscopy[10]. The effect should be stronger at the CBM according to DFT calculations because the atomic number of Pb ($Z = 82$), which constitutes the conduction orbitals, is much larger than Br ($Z = 35$). In addition, several experimental reports have also advocated the Rashba splitting effect by claiming bright triplet exciton emission at low temperatures for nanocrystals[36–38] and indirect bandgap formation[19,39,40]. Recently, Wang et al.[19] attributed a dual peak PL to the direct and indirect bandgap emission in MAPbI$_3$ films formed by the Rashba effect in the presence of MA cation breaking the centrosymmetry. Similar direct and indirect dual emission peaks are also present in

conventional indirect semiconductors such as Ge, GaSe, and transition metal dichalcogenide two-dimensional materials (MoS$_2$, MoSe$_2$), etc.[41,42]. The assignment of Peak 1 to direct emission and Peak 2 to phonon-assisted indirect emission is consistent with our experimental findings: (1) weak absorption/emission for Peak 2, (2) quadratic intensity dependence for both peaks, and (3) highly delocalized carriers existing in the band for Peak 2. Time-resolved PL (TRPL) studies provide strong evidence for the direct and indirect emissions as shown in Fig. 2e, f for high-quality (1 M) PC films and Supplementary Figure 6 for SCs. We employed the non-negative matrix factorization method to deconvolve the TRPL data at 165 K. The deconvolved effective lifetimes are around $110 \pm 10$ and $1540 \pm 50$ ns for Peak 1 and 2, corresponding to the recombination through the direct transitions and indirect transitions, respectively. At room temperature, both peaks show identical bi-exponential decay, indicating fast population exchange between the upper direct bands and lower indirect bands by phonon scattering.

We further confirmed the formation of the split spin valleys from the PL helicity that depends on the excitation light helicity at cryogenic temperatures. The technique is widely used to selectively excite spin valleys to achieve valley polarization, which provides a good indication of how well the valley identity of charge carriers is preserved before recombination[43,44]. The experimental schematic is shown in Fig. 3a. If there are spin-split bands due to the Rashba effect, we can selectively excite these bands using circular-polarized optical pumping. To exclude any instrumental polarization-dependent response, we kept the same detection polarization and only varied the incident polarization. Figure 3b displays typical right circularly-polarized ($\sigma_-$) PL spectra of MAPbBr$_3$ SC upon left ($\sigma_+$) and right ($\sigma_-$) circularly

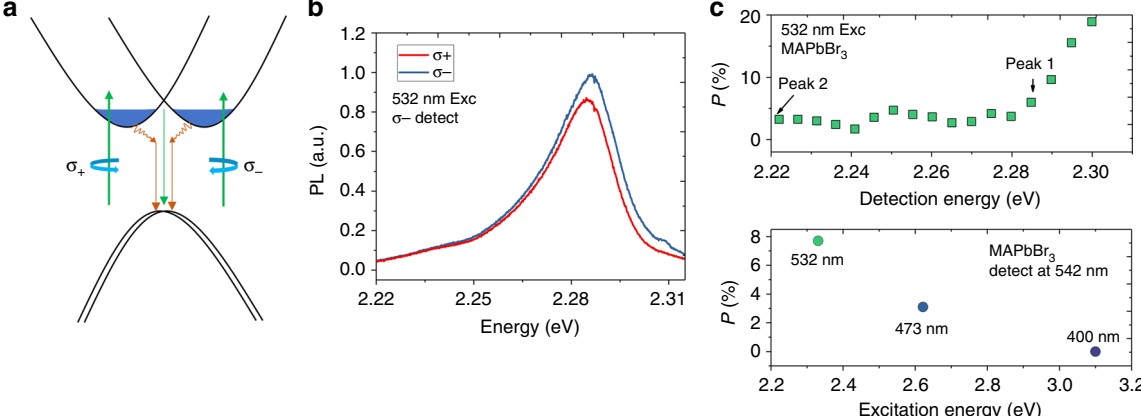

**Fig. 3** PL helicity upon circularly-polarized excitation. **a** Schematic of the Rashba split bands and selective excitation with circularly polarized light. **b** Right circularly polarized ($\sigma_-$) PL spectra of MAPbBr$_3$ SC upon left ($\sigma_+$) and right ($\sigma_-$) circularly polarized excitation with 532 nm laser at 77 K. **c** The degree of circular polarization (top panel) detected and (bottom panel) excited at different energies

polarized excitation with 532 nm laser at 77 K. It can be clearly observed that the PL helicity follows that of the optical pump, a signature of the optically pumped valley polarization. The same conclusion can be obtained when the detection polarization is reversed (Supplementary Figure 7). The degree of circular polarization of the PL is defined as[43]:

$$P = \left| \frac{I(\sigma_+) - I(\sigma_-)}{I(\sigma_+) + I(\sigma_-)} \right|, \qquad (1)$$

where $I(\sigma_+)$, $I(\sigma_-)$ are the PL intensity with left- and right-circular optical pumping, respectively. $P$ was found to be high near Peak 1 (around 8%), which however drops to only around 3% at Peak 2. The decrease of $P$ is a consequence of the dominance of the direct transitions at Peak 1 and the indirect transitions at Peak 2. The latter has a much slower rate compared to that of spin-flipping which smears out the polarization information. $P$ also drops significantly when the excitation wavelength is changed from near-resonance to off-resonance. For example, $P$ decreases to 3% and 0% with 473 and 400 nm optical pumping, respectively. This is attributed to the potential barrier formed between the two spin-split bands that preserves the initial photo-carrier spin with near-resonance excitation. Similar experiments were obtained for CsPbBr$_3$ SC (Supplementary Figure 7c–e). Recently, Niesner et al.[45] also observed that the photocurrent of MAPbI$_3$ SC is highly dependent on the light helicity when excited near the indirect bandgap, namely, the circular photogalvanic effect. Due to the strong SOC, the spin lifetime in lead halide perovskites is very fast (around ps)[46]. The observation of optical-pump polarization-dependent PL unambiguously point to the presence of spin-split bands near the band edge that extend the carrier spin lifetimes.

Nonetheless, the origin of the Rashba splitting in these lead halide perovskites remains vague. This is especially for CsPbBr$_3$, which possesses centrosymmetry at any temperature, where the Rashba splitting effect is not expected. To reveal the underlying mechanisms, the energy difference between Peak 1 and Peak 2 in the PL profiles is used as a gauge to evaluate the Rashba band-splitting effect. Figure 4a displays the temperature-dependent energy difference, in which the splitting of all the perovskite SCs remains constant at low temperature but starts to increase monotonically when the temperature is larger than around 60 K. The temperature dependence of the splitting effect follows a dependence on the phonon occupation, which is empirically

expressed as: $\Delta E = \Delta E_0 + A/(\exp(E_{ph}/k_B T) - 1)$, where $\Delta E_0$ is the temperature-independent term, $A/(\exp(E_{ph}/k_B T) - 1)$ is the temperature-dependent term. $E_{ph}$ is the energy of the phonon leading to the splitting effect, $k_B$ is the Boltzmann constant, and $A$ is the scaling factor. The fitting results are listed in Table 1. $\Delta E_0$ has a value of 20–40 meV in all of perovskite SCs studied leading to distinct dual emission peaks at low temperatures (Supplementary Figure 8) and is highly dependent on sample conditions. The temperature-independent term should arise from static centrosymmetry breaking by the non-spherical A-site organic cations (MA, FA), surface distortion or internal interface distortions (such as twinning of orthorhombic phase, inclusion of different phases)[32,34,45]. MAPbBr$_3$ generally possesses a higher static distortion compared to the other two by around 10 meV, which should be due to the centrosymmetry breaking by MA cations. Furthermore, it is interesting to note that without thermal distortion at low temperature, CsPbBr$_3$ SC exhibits a static splitting of around 26 meV, implying the contribution of surface/interface distortions is significant in causing Rashba splitting at low temperature. The fitted values of $E_{ph}$ are 12 to 18 meV, which corresponds well with the broadening of the PL peaks (Supplementary Figure 9), and is attributed to the Pb–Br longitudinal optical (LO) phonon modes (Pb–Br–Pb bending and Pb–Br stretchings)[25,47]. Hence, it is inferred that the temperature-dependent splitting is strongly related to the thermal-induced perovskite octahedra cage deformation. Since the Rashba effect can only occur when local centrosymmetry is broken and an electric field is present, which means, the thermal fluctuation of the octahedra cage at elevated temperatures must have a strong polar motif that yields intense local fields around the Pb atoms[32,35,48].

The recent work of Yaffe et al.[48] revealed the counter-intuitive notion that the "caged" atomic Cs cations within CsPbBr$_3$ perovskite is capable of generating a dipole, via short-lived (picosecond) structural/polar fluctuations. The implications for this are, of course, vast and their findings identify this to be an intrinsic physical feature of the anharmonic lead-halide perovskite lattice, i.e. the fluctuation vector direction is cation independent. Among its effects is a large zero-frequency Raman signature (sometimes termed quasi-elastic scattering), which manifests via a broad low-frequency continuum in an experimental Raman spectrum[49]. Fig. 4b presents selected Raman scattering spectra of CsPbBr$_3$ recorded between 300 and 80 K, down to a shift of around 22 cm$^{-1}$ from zero-frequency. At room temperature, there is a set of well-defined Lorentzian peaks

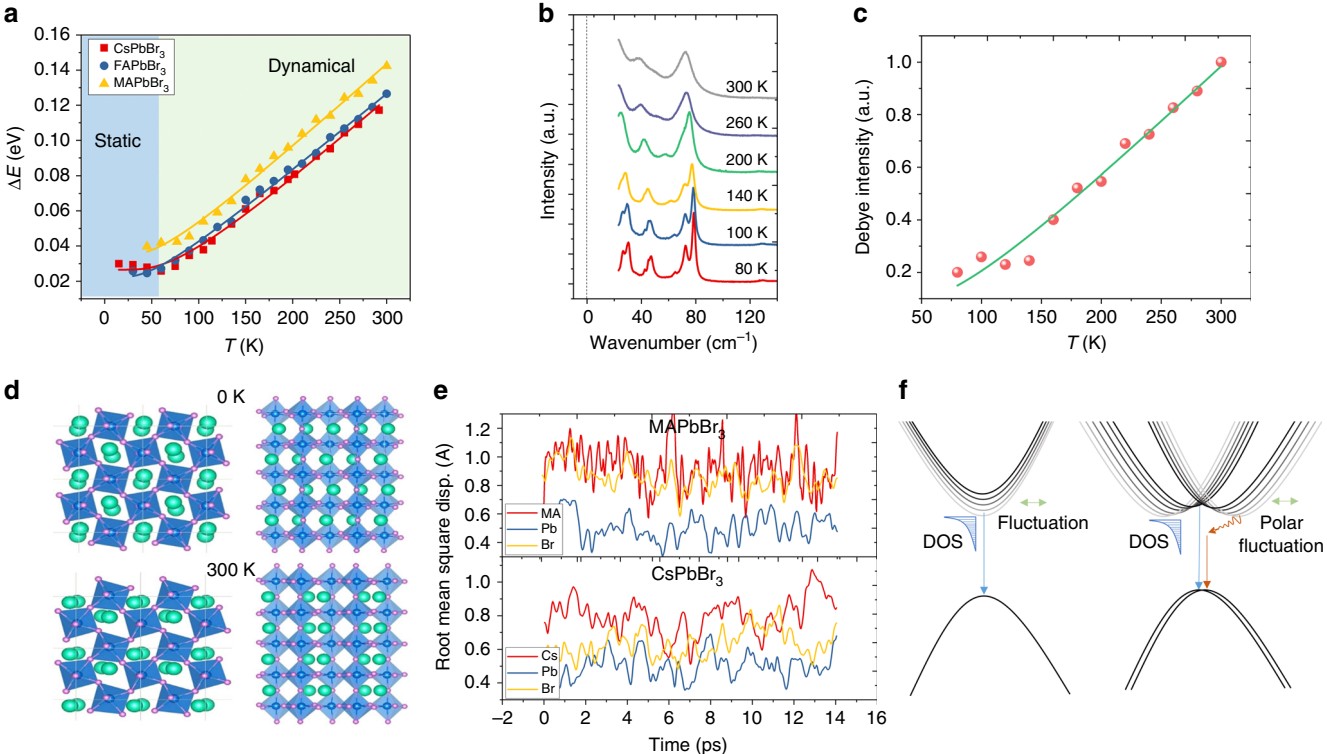

**Fig. 4** Temperature-dependent energy splitting in lead bromide perovskite SCs. **a** Temperature-dependent energy difference of Peak 1 and 2. **b** Temperature-dependent low-frequency Raman spectra of $CsPbBr_3$ SC. **c** Temperature-dependent relative intensity of the zero-mode extracted from **b**. Note that while the sudden increase in the intensity of the Debye contribution near 150 K is reproducible and appears to be real (i.e., while other phonon modes have a more regular evolution), the green line is merely a guide for the eye. **d** Typical instantaneous crystal structure of $CsPbBr_3$ at room temperature (300 K) (bottom panel). The crystal structure at 0 K is also shown for comparison (top panel). Left: top-view, right: side-view. Cyan: Cs, Blue: Pb, Purple: Br. **e** RMSDs of $MAPbBr_3$ and $CsPbBr_3$ atoms in cubic supercells simulated at 500 K. **f** Schematics of the formation of the Urbach tail states (left) and the indirect tail states (right). For clarity, only the tail states at the CBM are depicted

| Table 1 Extracted parameters from the temperature-dependent splitting curves | | | |
|---|---|---|---|
| Crystal | $\Delta E_0$ (meV) | $A$ (meV) | $E_{ph}$ (meV) |
| $CsPbBr_3$ | 26 ± 2 | 103 ± 17 | 18.6 ± 2.3 |
| $FAPbBr_3$ | 22 ± 2 | 61 ± 10 | 12.0 ± 1.7 |
| $MAPbBr_3$ | 35 ± 3 | 79 ± 20 | 14.1 ± 3.0 |

superimposed on a large broad signal which grows toward the un-shifted line, generated by local polar fluctuations (Supplementary Figure 10). As the thermal energy of the system is reduced, the octahedral vibrations become narrower, blueshift and their degeneracy are better resolved, in line with the classical interpretation of the temperature dependence of Raman active crystal modes. Importantly, the background generated by the zero-frequency band experiences a decline over this temperature range, becoming relatively small at 80 K. From the Raman data and the relevant literature, no phase change is experienced over these temperatures, only a further non-cubic distortion of the *Pbnm* phase. Figure 4b suggests that the local polar fluctuations at room temperature in an orthorhombic $CsPbBr_3$ structure are quite strong, and decline systematically, moving towards lower temperatures. To quantify the temperature dependence of local polar fluctuation strength, we adopt the same approach detailed by Yaffe et al., where a Debye relaxation model is used to fit the continuum underlying the low-energy Raman spectrum, centered on a Stoke shift of zero (see Supplementary Note 3 for details).

Tracking the relative intensity of the zero-mode for declining temperatures in Fig. 4c, we find that its contribution to the Raman spectrum decreases systematically, scaling with the occupation number of Pb–Br LO phonon modes. The consistency between the energy splitting (Fig. 4a) and the polar fluctuation strength (Fig. 4c), not only in thermal-driven trend but also in static versus dynamic distortion ratio, experimentally supports our proposal that the origin of the dual emission is from the Rashba splitting effect.

The results were further verified through MD simulations of the $CsPbBr_3$ lattice fluctuations, which we find to be characterized by large off-center displacements of the Cs ions and polar distortions of the $PbBr_6$ octahedra (Fig. 4d). The polar distortion of the $PbBr_6$ octahedra can then give rise to local electric fields extending beyond the characteristic Rashba length scale (1–2 nm), leading to a Rashba effect that fluctuates with lattice dynamics[32,35]. For example, the typical instantaneous structure as shown in Fig. 4d can be input into the DFT and the band structure shows strong Rashba splitting effect, in contrast to no splitting effect with the structure at zero temperature (Supplementary Figure 11). Note that the calculated instantaneous band structure does not mean an accurate (static) one at elevated temperatures, because the atoms are always vibrating. On electronic time scales, the electrons see a snapshot of the atoms spatially disordered due to their vibrational displacements, leading to dynamic changes in the band structure. This results in an average non-zero splitting effect and the existence of a density-of-states with indirect characteristics, residing below the direct band edge. Similar thermal polar distortion can exist in

MAPbBr$_3$ and FAPbBr$_3$, which share the same PbBr$_6$ octahedral cage. Although the inorganic and organic A-site cations have different vibrational characteristics, our findings show that they do not play an important role in determining the temperature-dependent Rashba splitting. Their phonon frequencies are not coupled to the Pb–Br frequencies to collectively influence the temperature trend of the splitting. To further validate this, we compared the root-mean-square displacements:

$$\text{RMSD} = \sqrt{\frac{1}{N}\sum_{i=1}^{N}(r_i - r_{0i})^2}$$ where $N$ is the number of the equivalent atoms, $r_i - r_{0i}$ is the displacement of the $i$th atom from its equilibrium position $r_{0i}$. The time-dependent trends of each atom in MAPbBr$_3$ and CsPbBr$_3$ cubic supercells are shown in Fig. 4e within 15 ps. The RMSDs at 500 K for MAPbBr$_3$ atoms are (MA: 0.95 Å, Pb: 0.51 Å, Br: 0.86 Å), and for CsPbBr$_3$ atoms are (Cs: 0.74 Å, Pb: 0.54 Å, Br: 0.65 Å). This indicates the change of A-site cations (i.e., MA, FA, and Cs) here does not affect the Pb–Br displacements much. It would then be reasonable to conclude that any phase change directly linked to the motional freedom of A-site cation should also not play a role in the splitting effect. Note that the indirect tail states' formation mechanism is similar to the formation of an Urbach tail: lattice thermal fluctuation perturbs the Hamiltonian through changing the potential or creating local fields that depend on the RMSD from equilibrium[50]. It forms tail states with an exponential distribution below the bandgap (Fig. 4f, left). The difference is that, in lead halide perovskites, the thermal fluctuation with prominent polar influences, together with the large SOC of lead, splits the degeneracy of the CBM, forming spin- or momentum-mismatched tail states (Fig. 4f, right). Amidst the review period of this manuscript, two recent theoretical reports on the calculated dynamical Rashba splitting energy for CsPbI$_3$ came to our attention[51,52]. At room temperature, the calculated dynamical Rashba split is around 10–20 meV, which is approximately one order smaller than our experimental results. We attribute this difference to the more significant atomic vibrations of the lighter bromine compared to iodine. Furthermore, imperfect lattice, polaron formation, and phonons involved in the indirect emission processes as well as the approximations in the calculation methods may also account for the difference. Nonetheless, these theoretical reports lend crucial support for the dynamical Rashba effect.

## Discussion

A semiconductor's fundamental optical absorption edge underpins the operation of a wide range of optoelectronic devices. Thermal-induced lattice fluctuation is an important contribution to the formation of the fundamental optical absorption edge in many conventional semiconductors. Presently, its exact mechanism remains under intense debate, with proposed theories such as deformation-potential theory, Stark-shift exciton mechanism, electric-field ionization of the exciton, phonon-side band, Franz–Keldysh effects, etc.[50]. The family of semiconducting halide perovskites with the soft nature of Pb-halide bonds is expected to be influenced by this effect. There are only a few reports on the Urbach tail characteristics of lead halide perovskites[53,54]. Deep insights into its influence on the optoelectronic properties remain lacking. Specifically, our experimental results on lead-bromide SCs and high-quality PC films with various A-site cations reveal unique optical absorption and emission properties driven by thermal-induced lattice fluctuation, which cannot be solely explained by the reabsorption effect from the Urbach tail. These results strongly support the indirect transition nature of the subgap tail states mediated by thermal-induced lattice polar fluctuation in all three lead bromide perovskites: CsPbBr$_3$, FAPbBr$_3$, and MAPbBr$_3$, regardless of the nature of the A-site

cations. Strong local fields may be generated by the PbBr$_6$ octahedral cage polar deformation at elevated temperature, which leads to the dynamical Rashba splitting and consequently the formation of these tail states distinct from the absorption edge (or Urbach tail) formation in typical semiconductors. These findings have a profound bearing on our understanding of the photophysics of halide perovskites and have strong implications on their light emission and light-harvesting applications.

Let us first examine the implications for light emission applications. The presence of thermal polar fluctuations and their associated indirect tail states can lead to the dual emission phenomena that were not observed in conventional polar semiconductors with an Urbach tail. Strictly, the second emission peak at elevated temperature with thermal fluctuation should correspond to the energy position when the product of the density of the carriers and the transition probability reaches its maximum. It is reasonable that the low-lying indirect emission peak can be as prominent as the direct one when more carriers reside at the indirect tail than in the direct band, and the phonon-mediated momentum mismatch is relatively small, i.e., relatively strong Fröhlich coupling may also assist the crystal momentum transfer in an indirect emission process[25,55].

The appearance of the dual emission would degrade the color purity of the emitted light. A large fraction of the carrier population is dispersed in the indirect tail with relatively lower transition probability. This points to difficulties in obtaining high-efficiency light emission and lasing from lead halide perovskites, at least for high-quality SCs or large grain PCs. However, the soft nature of Pb-halide bonds makes the perovskite lattice prone to deformation by the surface, disorder/defects, etc.[35], which disrupts the long-range ordering and changes the direct and indirect characteristics of lead halide perovskites. Likewise, the radiative recombination rate can be significantly altered with varied defect doping, electron–phonon coupling, or quantum confinement. These factors can complicate the emission properties that lead to non-prominent or prominent indirect/direct emissions. If we treat the indirect tail as an indirect bandgap semiconductor with its average bandgap position located at Peak 2, the electron–hole recombination coefficient at the direct band edge of an indirect bandgap materials can be expressed as[56]:

$$B_1 = \frac{(2\pi)^{1/2}he^2n}{M_c c^3 m_0^{5/2}(k_B T)^{3/2}}\left(\frac{m_0}{m_e + m_h}\right)^{3/2} \\ \left(1 + \frac{m_0}{m_e} + \frac{m_0}{m_h}\right)E_{gd}^2 \exp\left(\frac{E_{gi} - E_{gd}}{k_B T}\right), \tag{2}$$

$E_{gi}$ ($E_{gd}$) are the indirect (direct) bandgaps, $m_e$($m_h$) are the electron (hole) effective masses, $m_0$ is the electron mass, $h$ is Planck constant, $k_B$ is Boltzmann constant, $T$ is the temperature, $e$ is the elementary charge, $c$ is the light speed in vacuum, $n$ is the refractive index, and $M_c$ is the number of equivalent minima in the conduction band. The calculated $B_1$ is sensitive to the exponential term $\exp((E_{gi} - E_{gd})/k_B T)$ determined by the direct-indirect bandgap difference, and therefore can be significantly changed with different surface distortion and defect density. As shown in Fig. 5a, we observed a striking difference in the effective e–h recombination coefficients between PC TFs and SCs (details of the fitting are shown/provided in Supplementary Figure 12 and Supplementary Note 4). At room temperature, the effective e–h recombination coefficient of FAPbBr$_3$ TFs approaches $10^{-8}$ cm$^3$ s$^{-1}$, which is three orders higher than that of the bulk of the SCs. Similarly, the three orders difference in electron–hole recombination rate coefficients ($10^{-11}$ to $10^{-8}$ cm$^3$ s$^{-1}$) for different sized perovskite crystals were also reported (Supplementary Table 1). In Fig. 5b, we observed the PL intensity is inversely proportional to the energy

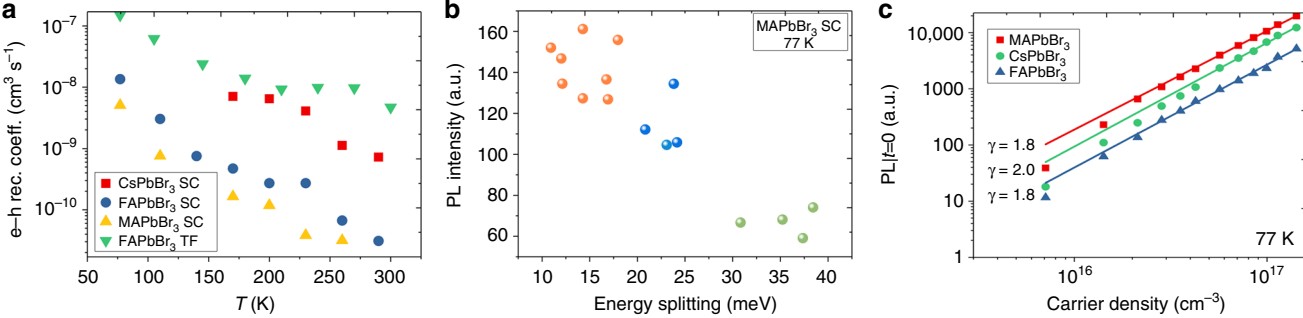

**Fig. 5** Implications for light emitting and light harvesting applications. **a** Temperature-dependent effective e–h recombination coefficients for different SCs and PCs. **b** The PL intensity versus the Rashba band-splitting for MAPbBr$_3$ SCs. The points were collected with three different MAPbBr$_3$ crystals (depicted by three different colored dots) at different sampling points. **c** The initial PL intensity under different excitation density at 77 K for lead bromide perovskite SCs showing a quadratic dependence on excitation density in the range of $10^{16}$–$10^{17}$ cm$^{-3}$, indicating the dominance of the free carrier band edge emission

splitting at low temperature (77 K). Note that the SCs used in these measurements were prepared under the same conditions. Different surface distortions among different crystals and points may account for the different Rashba splitting effect and PL properties at low temperature. These results suggest the ease of lattice distortion and the associated change of the indirect tail states play a significant role in the emissive properties (PLQY, dual peaks, etc.) of lead halide perovskites. The presence of the indirect tail states also implies that it is difficult for high-quality large SCs to function as lasing media at room temperature. This is consistent with our observations that all our SCs do not show any amplified spontaneous emission at room temperature even when the carrier concentration reaches $10^{20}$ cm$^{-3}$ (Supplementary Figure 13).

We next examine the implications for light-harvesting applications. The existence of an indirect bandgap slightly below the direct transition edge can also help to prolong carrier lifetimes[32,33,39]. This ensures long carrier diffusion lengths favorable for charge extraction in a solar cell with relatively low carrier mobility. On the other hand, for high and balanced carrier mobilities such as in optimized lead halide perovskites, it may not help to increase the ultimate efficiency in the radiative limit[57]. We would like to highlight that exciton dissociation efficiency could also possibly be improved with the presence of thermal polar fluctuation. Given a 30 meV exciton binding energy and approximately 0.12 $m_e$ reduced effective mass for carriers measured previously[58], the free carrier population is less than 20% at 77 K for a carrier concentration in the range of $10^{16}$–$10^{17}$ cm$^{-3}$ according to Saha ionization equation[59]; dominant exciton emission is therefore expected at liquid nitrogen temperature (77 K). However, we found that the free carrier emission still dominates for all the lead bromide perovskite SCs at such low temperatures (Fig. 5c). As a comparison, CdS SC with similar bandgap (2.42 eV) and exciton binding energy (28 meV)[60] shows exciton emission even at room temperature (Supplementary Figure 14). Surface band-bending may form a surface depletion region that assists electron–hole dissociation by the built-in electric field[61]. However, we can exclude this effect from the similar observations of the carrier generation mechanism with two-photon excitation which mainly excites the bulk part of the crystals (Supplementary Figure 15). We interpret that the photogenerated charges are rapidly separated due to electrostatic potential fluctuation coupled to the inorganic lattice dynamics. The organic cations which were believed to be critical in causing potential fluctuation and charge separations previously[62,63] may not be essential as H. Uratani and K. Yamashita[64] demonstrated through MD simulations. This will account for the quadratic dependence on injection carrier density at low temperatures for

SCs in our experiments. Again, the susceptibility of the lattice fluctuation to a local environment may help to explain a broad distribution of the reported exciton-binding energies.

In summary, through systematic time-dependent and temperature-dependent spectroscopy studies in a series of lead bromide SCs with different cations (Cs, FA, MA) correlated with DFT and MD calculations and electrical characterizations, we reveal the existence of a weak transition in the absorption and emission spectra below the direct band edge in all the three SCs— attributed to the tail states with indirect transition characteristics arising from static and thermal-induced dynamic lattice polar distortions. The PbBr$_6$ octahedral polar distortion provides a local electric field for the Rashba splitting that results in the formation of states with slight momentum mismatch below the direct bandgap. This dynamical Rashba splitting effect at elevated temperature is distinct from the static Rashba splitting effect caused by non-spherical A-site cations or surface induced lattice distortion. The thermal polar fluctuations and their associated indirect tail states in lead halide perovskite SCs result in low photoluminescence quantum yield and difficulties in realizing lasing, long carrier lifetimes, and efficient charge separation even under low temperature. Our findings rationalize the structure–function relationship of lead halide perovskites for optoelectronic applications and may have far-ranging implications for the design of emergent halide perovskites.

## Methods

**Single crystal preparation.** Cesium bromide (CsBr), lead bromide (PbBr$_2$), dimethylformamide (DMF), acetonitrile (ACN), dimethyl sulfoxide (DMSO) (≥99.9%), and γ-butyrolactone (GBL) (≥99%) were purchased from Sigma-Aldrich. All chemicals were used as received, without any further purification. Formamidinium bromide (FABr) and MABr were synthesized using previously reported methods[65]. FAPbBr$_3$ and MAPbBr$_3$ SCs were prepared using the reported inverse temperature crystallization method[66]. Briefly, 1 M perovskite precursor solutions, FAPbBr$_3$, and MAPbBr$_3$ in DMF were kept at 75 °C on a hotplate for several hours to allow SCs to form and grow. CsPbBr$_3$ SCs were prepared with a modified method from literature[67]. Briefly, sub-mm sized CsPbBr$_3$ seeds were first prepared by heating an over-saturated precursor solution (with equal molar amounts of PbBr$_2$ and CsBr solids at the bottom of a 20 mL glass vial) on a hotplate at 75 °C. Small crystals attaching to the sidewall of the glass vial can be collected after 48 h. Right after collection with tweezers, several seeds were positioned into the growth solution for further growth. Briefly, methanol was added dropwise into a 0.45 M CsBr and PbBr$_2$ precursor DMSO solution at room temperature until a saturated precursor solution was achieved. The resulting solution was then filtered with a PTFE filter (0.2 μm). Several seeds were then added to the filtered growth solution. The growth solution with seeds was immediately positioned onto a hotplate at 30 °C for another 48 h. All crystals were washed with an excess amount of ACN after collection. After drying the crystals with Kimwipes, they were finally annealed at 50 °C for 5 min to remove any residual solvent on the crystal surface. The crystals have a typical dimension of 2–3 mm × 2–3 mm × 0.5–1 mm.

**Photoluminescence and time-resolved photoluminescence spectroscopies**. (Time-integrated) photoluminescence measurements were conducted by directing the excitation laser pulses to SCs or thin films. The laser pulses were generated by a 1-kHz regenerative amplifier (Coherent Libra, 800 nm, 50 fs, 4 mJ). A mode-locked Ti-sapphire oscillator (Coherent Vitesse, 80 MHz) was used for the amplifier. For 800 nm pump, the laser from the regenerative amplifier was directly used. For 400 nm, a BBO crystal was used to double the frequency of 800 nm. The photoluminescence was collected at a backscattering angle by a spectrometer (Acton, Spectra Pro 2500i) and CCD (Princeton Instruments, Pixis 400B). For circular-polarized pump and circular-polarized PL measurements, a quarter-waveplate and polarizer were placed in the path of the excitation beam and another pair of quarter-waveplate and polarizer were placed at the PL detection path. Time-resolved photoluminescence was collected using a streak camera system (Optronis Optoscope), which has an ultimate temporal resolution of 6 ps. For the temperature-dependent PL and TRPL measurements, the crystals were adhered to the cryostat (Janis) copper sample mount with silver paste. The temperature was cooled down using liquid nitrogen.

**Diffuse reflectance spectroscopy**. Diffuse reflectance spectra at room temperature for MAPbBr$_3$ SCs were obtained using an integrating sphere coupled to the commercial UV-Vis-NIR spectrophotometer (Shimadzu UV-3600). The temperature-dependent diffuse reflectance was measured using a Nikon microscope equipped with a liquid-nitrogen cooled cryostat (Janis). The white light from a tungsten bulb was directed onto the SC with ×10 objective and the diffused light was collected with the dark field mode. The incident light intensity was calibrated by measuring the reflected light intensity from a quartz substrate and calculated with the known quartz refractive indices. Later the diffused light from the SCs was collected and divided by the incident intensity to obtain the diffuse reflectance spectra. The measured data are consistent with the commercial one at room temperature.

**Raman scattering spectroscopy**. The infrared Raman measurement (808 nm) was carried out using a XY-Dilor spectrometer associated with a Ti-Sapphire (Ti: Al$_2$O$_3$) tunable laser which was pumped by an Argon laser. At this wavelength, the sample is transparent and no heating effect has been observed. We have used a long working distance ×40 objective with a numerical aperture of 0.4 and selected a laser power under objective of 7 mW. A Princeton Instruments (PI) micro-cryostat able to monitor the sample temperature in the 80–300 K range has been used.

**EQE measurements**. EQE measurements were performed on a 0.5-mm-thick MAPbBr$_3$ SC working as a photodetector. Both sides of the SC were evaporated with Au electrodes, with one side electrode semi-transparent (around 20-nm-thick Au) to let incident light pass through. The performance was measured with 40 nW cm$^{-2}$ light intensity and 4 V bias.

**DFT and MD calculations**. We employ the all-electron-like projector augmented wave method[68] and the Perdew–Burke–Ernzerhof exchange-correlation potential[69] as implemented in the VASP code[70]. The semi-core of Cs (5s and 5p orbitals) and Pb atoms (5d orbital) is treated as valence electrons, i.e., nine valence electrons for Cs ($5s^25p^66s^1$) atom and 14 valence electrons for Pb ($5d^{10}6s^26p^2$) atom. In optimizing calculation, the cut-off energy for the plane wave expansion of the wave functions is 500 eV. The Hellmann–Feynman forces are less than 0.01 eV Å$^{-1}$. The $4 \times 4 \times 3$ Monkhorst-Pack grid of $k$-points for Brillouin zone integration of CsPbBr$_3$ was used in the calculations of structural optimization. The simulations of MD based on the DFT of CsPbBr$_3$ were performed at 300 K and 500 K. In the calculations, we employed a $2 \times 2 \times 2$ supercell. The Tkatchenko–Scheffler scheme has also been employed. Based on the relaxed structures at 0 K, the supercells were heated up to 500 K. After that, a 15 ps MD simulation was carried out in the canonical ensemble with a 1.5 fs time step.

## Data availability
The data that support the findings of this study are available from the corresponding authors upon request.

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

## Acknowledgements

Financial support from Nanyang Technological University start-up grant M4080514; the Ministry of Education AcRF Tier 1 grants RG101/15 and RG173/16 and Tier 2 grants MOE2015-T2-2-015 and MOE2016-T2-1-034; the NTU-A*STAR Silicon Technologies Center of Excellence Program Grant 11235100003; the US Office of Naval Research (ONRGNICOP-N62909-17-1-2155) and from the Singapore National Research Foundation (Programs NRF-CRP14-2014-03 and NRF2018-ITC001-001) is gratefully acknowledged. M.B.J.R. and J.H. acknowledge financial support from the Research Foundation-Flanders (ZW15_09-GOH6316, AKUL/15/15-G0H0816N, G.098319N), KU Leuven Research Fund (C14/15/053), the Flemish government through long-term structural funding Methusalem (CASAS2, Meth/15/04) and the Hercules foundation (HER/11/14). H.Y. and J.A.S. acknowledge the Research Foundation-Flanders (FWO) for postdoctoral fellowships. B.W. acknowledges the research conducted at NTU and the support from SCNU during the final manuscript revisions.

## Author contributions

T.C.S. and B.W. conceived the idea for the manuscript and designed the experiments. B.W. developed the basic concepts and conducted the spectroscopic characterization. J.H. and H.Y. fabricated the single crystals and performed the physical characterization. J.A.S., M.B.J.R., and P.P. performed the temperature-dependent Raman scattering measurements. D.G. and B.W. performed the polarization-dependent PL experiments and analysis. Q.X. performed the DFT and MD calculations. J.F., Y.F.N., N.F.J., A.S., M.G., S.M., and N.M. contributed to the thin film fabrication and characterization. All authors contributed to the data analysis and writing the manuscript. T.C.S. and J.H. guided the experiments, discussed the data, and led the project.

## Additional information

**Competing interests:** The authors declare no competing interests.

