## [Peer Review File · Nature Communications]

Reviewers' comments:

Reviewer #1 (Remarks to the Author):

In this work the authors carried temperature dependent PL studies on metal halide bromide crystals. Independent of the cation ion they find 2 emission peaks. Both peaks originate from bimolecular processes. The energy difference between the lower and higher energy peak increases with temperature, again more or less independent of the cation. Thin films show only a single emission close to the high energy peak of the single crystal. The authors try to interpret these data by employing the Rashba model. This model, basically explained in figure 4e has been used in previous theoretical and experimental studies to explain results on metal halide perovskites. As such the present work is not very new. As claimed in the present abstract by the authors, this model is also able to explain all PL observations. Unfortunately not much evidence is given why Rashba splitting is the mechanism explaining their results. Therefore I cannot advise publication in its present form. Below I summarize some doubts I have regarding their interpretation.

- Comparison the PL of Cs, MA and FA perovskite is a very logical choice in view of the huge differences in the cation motion. In Figure 3b they show ΔE as function of the temperature. One would not expect that these dependencies would follow the same trends for the different cations. Apart from different motional freedom of the cation also phase changes (possibly via cation motion) would have profound effects on the splitting. No explanation is given here.
- Various causes are presented for the Rashba effect: including cation motion, thermal lattice distortion but also surface defects. In view of the fact that the authors measured films and crystals for perovskites with different cations, it would be logical that some of these possible causes could be omitted. However this is not discussed at all.
- A similar unspecified remark is mentioned in the abstract line 28 (and not mentioned at other places): indirect 'Urbach-like' states. What are those states?
- Seems that something is wrong of the color coding of Figure 2F. In the present case not much difference is found for the two emission peaks, while in the text they mention there are huge changes.
- In the orthorhombic phase of MAPbBr_3 a huge broad emission peak is observable above 600 nm, which is unfortunately cut off, which is probably related to radiative process 6 in Figure 4e.

Reviewer #2 (Remarks to the Author):

The authors describe spectroscopic measurements of bromide-based lead halide perovskite single crystals with different organic cations (Cs, MA, FA) and study in detail the sub-bandgap states. They vary the temperature, excitation density and wavelength, and morphology, and find that the below bandgap feature present in PL and absorption is linked to the indirect transition generated via the Rashba effect.

Generally, the data is interesting and the interpretation in many places makes sense. However, the writing is very bad in some places, and the paper is oversold in others. Also, many findings are not as novel as the authors claim they are. Thus, while the paper should be published, and might be suitable for Nature Communications, major revisions are necessary, both in writing and interpretation. My main concern is the claim of novelty. The data quite nicely shows the Rashba effect, but all the effects have been observed in similar systems with similar materials. E.g., as the authors acknowledge, the low-energy peak in the PL has been assigned to the indirect gap before (in MAPI, DOI: 10.1039/C6EE03474H), the low-temperature dependence has been studied before (in MAPI, doi:10.1038/nmat4765), the Rashba effect has been observed in bromide perovskites by ARPES (/doi.org/10.1103/PhysRevLett.117.126401) etc.; Many of the claims are not new, the main novelty of the paper in my view is the fact that it compares different cations, and single crystals to thin films.

The authors should tone down the paper and work out exactly what is novel about their study. This would also mean that the big claims about the far reaching impact of their findings would need to be toned down.

Additional detailed comments that need addressing:

- 1) The abstract and the beginning are written terribly, like a sales-piece. Phrases like “enormous potential for various impactful deep tech applications” have no place in a serious scientific article. There are many of such phrases and buzzwords, and also many spelling mistakes. Also terms like “giant SOC” suggest technical terms, but here giant is presumably just used as synonym for large.
- 2) On p.3, when discussing the various differences between SCs and TFs, the authors should cite works for each of the examples.
- 3) In several places do the authors claim that the indirect bandgap would be beneficial for photovoltaics. This is only true if the devices are limited by the mobility of the charge carriers. As shown by Rau & Kircharz (DOI: 10.1021/acs.jpcclett.7b00236), this is not the case for perovskites, where a completely direct bandgap would lead to higher efficiency.
- 4) When discussing Figure 1d, the authors do not give a reason as to why they do not see the low-energy peak in the thin films. Presumably, the absorption is just too weak?
- 5) For the same figure, why is the EQE shape so strange? I.e, why is there a peak right at the bandedge?
- 6) Far too little detail is given about the experimental details. Basic measures, e.g. how big are the single crystals, what is the difference in the two crystals shown in the SI for each material, what is the geometry of the photodetector etc. are omitted.
- 7) One of the nicest findings of the paper is the difference in the dynamics of the two peaks. The indirect peak has slower dynamics at low T and identical dynamics at RT. Can the authors expand on the interpretation, for example calculate how much of the emission they expect to be in the direct/indirect peak at a certain T, from the energy difference and the phonon DOS.
- 8) From the ARPES measurements on MAPbBr₃ one would suggest the Rashba splitting to reside mainly in the valence band (doi.org/10.1103/PhysRevLett.117.126401), yet from the DFT calculations here, and from the schematic figures, the authors suggest that the conduction band is more split. Can you explain that discrepancy?
- 9) In Figure 4 b & c the authors compare different single crystals, but never tell us what the difference is!
- 10) Towards the end of the paper the authors speculate on the indirect gap to be beneficial for exciton dissociation. It is not clear to me how their interpretation makes sense. They claim that because the electron relaxes into the indirect CBM. How would that help to dissociate the exciton? Only if one changes the delocalization in real space would the binding energy be reduced, and I do not see how this could be done by thermalization into the CBM.

Reviewer #3 (Remarks to the Author):

Please find attached file.

Reviewer #4 (Remarks to the Author):

In NCOMMS-18-05870 the authors report an experimental, optical spectroscopic study on halide perovskites where the interplay between thermal atomic displacements and electronic properties is investigated.

By means of temperature dependent photoluminescence and reflection measurements, the authors reach the conclusion that strong spin-orbit coupling combined with symmetry breaking due to thermal

fluctuations cause tail states in the conduction band (Due to a Rashba effect). These tail states are expressed as a dual peak in the photoluminescence spectra.

The study is comprehensive and has merit. The findings are important and deserve publication. Yet, there are several issues that must be resolved prior to publication:

1. The authors claim that the dual peaks in the photoluminescence are intrinsic to the crystal. Though they provide some evidence for their claim, there is evidence to the contrary. In DOI:10.1021/acsnano.6b02734, Tilchin et al report the temperature dependent PL and absorption of MAPbBr₃ single crystals. In their data, the low energy peak is considerably less pronounced than here (compared to figure 3A at 45 K). The aforementioned paper is not cited by the authors (I am sure it was not ignored but missed). I am also aware of other unreported data (presented in conferences) where the low energy peak is even less pronounced. I wonder how this fact coincides with the authors' interpretation. This point must be addressed in the text.

2. I am not sure why the authors suggest that the Rashba splitting (that, the best of my understanding, is responsible for the low energy peak) occurs exclusively in the conduction band. To the best of my experimentally measured (DOI:10.1103/PhysRevLett.117.126401) for the valence band.

3. Moreover, the Rashba splitting should be symmetrical around the gamma point (unlike what is depicted in figure 3d and 4e) and as the authors point out, creates an indirect band gap. On this point two issues are not clear to me:

a. How come the photoluminescence from the indirect gap is so efficient that it has the same order of magnitude of the direct band gap photoluminescence?

b. If low energy photoluminescence does indeed result from thermally induced Rashba splitting, shouldn't one expect an overlay of a distribution of split bands - meaning effectively broadened bands eventually - instead of a clear discrete split band?

Reviewer #1 (Remarks to the Author):

In this work the authors carried temperature dependent PL studies on metal halide bromide crystals. Independent of the cation ion they find 2 emission peaks. Both peaks originate from bimolecular processes. The energy difference between the lower and higher energy peak increases with temperature, again more or less independent of the cation. Thin films show only a single emission close to the high energy peak of the single crystal. The authors try to interpret these data by employing the Rashba model. This model, basically explained in figure 4e has been used in previous theoretical and experimental studies to explain results on metal halide perovskites. As such the present work is not very new. As claimed in the present abstract by the authors, this model is also able to explain all PL observations. Unfortunately not much evidence is given why Rashba splitting is the mechanism explaining their results. Therefore I cannot advise publication in its present form.

Below I summarize some doubts I have regarding their interpretation.

Response: We thank the reviewer for his valuable time and comments. We acknowledge that the significance and novelty of the findings in our original manuscript may not have been adequately highlighted, which could have affected his views. Indeed, Rashba splitting have been theoretically and experimentally observed in several reports as pointed out by the reviewer. However, detailed understanding of the origin(s) of the splitting effect is still lacking with only a few experimental reports. Most reports simply attributed the Rashba splitting to arise from the breaking of the centro-symmetry of the lattice by the A-site cation. In our work, we uncover new insights into the origins of the Rashba effect in lead halide perovskites. Firstly, we discerned the static effects from organic cations and surface effect. Most importantly, we found that the thermal induced polar fluctuation is the main cause of the Rashba splitting at room temperature. Consequently, unlike the Urbach tail states in conventional polar semiconductors, the indirect nature of the tail states of lead halide perovskites give rise to novel optical properties like dual emission peaks. To further confirm the thermal polar fluctuation and indirect tail states formation, we have also performed new studies involving temperature-dependent low-frequency Raman scattering measurements and correlate with molecular dynamics (MD) simulations. We now have direct evidence of the thermal polar fluctuation and its relationship with the dual emission. We are therefore confident that the quality and novelty of our revised manuscript will meet the stringent standards of *Nature Communications*.

(1) Comparison the PL of Cs, Ma and Fa perovskite is a very logical choice in view of the huge differences in the cation motion. In Figure 3b they show ΔE as function of the temperature. One would not expect that these dependencies would follow the same trends for the different cations. Apart from different motional freedom of the cation also phase changes (possibly via cation motion) would have profound effects on the splitting. No explanation is given here.

Response: We thank the reviewer for his comments and important suggestions. The band structures near the band edges are dominated by PbX_6 octahedra, *i.e.*, the CBM comprises of the Pb p orbital and the VBM is the anti-bonding state of Pb s and X p orbitals. The energy levels of MA, FA, and Cs is far away from the band edges. Hence, it is expected that changing the A-site cations will not significantly alter the

band edge properties and their dependence on temperature. The temperature-dependent trends of the ΔE in lead-bromide perovskites reflect the thermal deformation of the PbBr_6 octahedra, which is directly related to the occupation number of Pb-Br phonons.

One may expect the A-site cation motion can efficiently couple to PbBr_6 octahedra deformation, and may work indirectly on the observed temperature-dependence of the ΔE . However, from Fig 3a of the main manuscript, the dependence of ΔE with temperature remains the same for CsPbBr_3 , MAPbBr_3 and FAPbBr_3 systems. This indicates that the frequencies of Cs, MA or FA phonons coupled to Pb-Br modes do not change ΔE significantly. This is consistent to that only the Pb-Br LO phonon modes contribute (via Fröhlich coupling) to the electron-phonon scattering and PL broadening in 3D perovskites (*Nat. Comm.*, 7:11755, (2016)), rather than through a mixing of A-site cation phonon modes with the Pb-Br modes.

To validate this further, we performed molecular dynamics simulations to obtain the atomic root-mean-square displacements: $\text{RMSD} = \sqrt{\frac{1}{N} \sum_{i=1}^N (r_i - r_{0i})^2}$ where N is the number of the equivalent atoms, $r_i - r_{0i}$

is the displacement of the i^{th} atom from its equilibrium position r_{0i} , of MAPbBr_3 and CsPbBr_3 in cubic phase at 500 K as shown below in **Figure R1**. The RMSDs at 500 K for MAPbBr_3 atoms are (MA: 0.95 Å, Pb: 0.51 Å, Br: 0.86 Å), and for CsPbBr_3 atoms are (Cs: 0.74 Å, Pb: 0.54 Å, Br: 0.65 Å). Furthermore, a few recently published theoretical works also support our findings here. For example, Uratani and Yamashita compared the molecular dynamics of CsPbI_3 , FAPbI_3 and MAPbI_3 , in which they observed similar lattice fluctuation and efficient charge separation (*J. Phys. Chem. C* 2017, 121, 26648–26654). McKechnie *et al* predicted similar spin splitting effect in CsPbI_3 and MAPbI_3 (arXiv:1711.00533v1). We therefore believe that the change of A-site cation from Cs to MA will not significantly change the PbBr_6 octahedra dynamics.

Figure R1. The time-dependent root-mean-square atomic displacements of MAPbBr_3 and CsPbBr_3 with MD simulations.

With regards to the abovementioned points, it is reasonable to infer that the phase changes that are linked the motional freedom of the A-site cations will not play an important role in determining the

temperature dependent ΔE in the 3D perovskites studied here. Even though we observed abrupt band gap changes of both Peak 1 and Peak 2 positions near phase-transition points as shown in Figure 1e. However, their energy difference ΔE is still much affected, as it is only directly related to the PbBr_6 deformation as discussed above.

In response, we added discussion in Page 11, Line 9 from the bottom.

“Although the inorganic and organic A-site cations have different vibrational characteristics, our findings show that they do not play an important role in determining the temperature-dependent Rashba splitting. Their phonon frequencies are not coupled to the Pb-Br frequencies to collectively influence the temperature trend of the splitting. To further validate this, we compared the root-mean-square displacements: $\text{RMSD} = \sqrt{\frac{1}{N} \sum_{i=1}^N (r_i - r_{0i})^2}$ where N is the number of the equivalent atoms, $r_i - r_{0i}$ is the displacement of the i_{th} atom from its equilibrium position r_{0i} . The time-dependent trends of each atoms in MAPbBr_3 and CsPbBr_3 cubic supercells are shown **Figure 3e** within 15 ps. The RMSDs at 500 K for MAPbBr_3 atoms are (MA: 0.95 Å, Pb: 0.51 Å, Br: 0.86 Å), and for CsPbBr_3 atoms are (Cs: 0.74 Å, Pb: 0.54 Å, Br: 0.65 Å). This indicates the change of A-site cations (*i.e.*, MA, FA and Cs) here does not affect the Pb-Br displacements much. It would then be reasonable to conclude that any phase changes that are directly linked to the motional freedom of A-site cations should also not play a role in the splitting effect. Note the...”

(2) Various causes are presented for the Rashba effect: including cation motion, thermal lattice distortion but also surface defects. In view of the fact that the authors measured films and crystals for perovskites with different cations, it would be logical that some of these possible causes could be omitted. However this is not discussed at all.

Response: The Rashba splitting originates from the centro-symmetry breaking of the perovskite lattice. These include (a) static centro-symmetry breaking, e.g., by the non-spherical A-site cation, surface distortion and (b) dynamical symmetry breaking as we discussed in the manuscript by atomic vibration. In the manuscript, we observed in real crystals (including the polycrystalline films and single crystals), none of these causes can be omitted. For example, for the symmetric CsPbBr_3 SC at low temperature when thermal effect is not involved, the splitting effect is still non-zero, this implies the surface effect is present (26 meV for the SC used in the fitting). In addition, the effect of A-site organic cation is obvious, as we always observe a higher value of the splitting effect for MAPbBr_3 compared to CsPbBr_3 .

In response to the reviewer, we made a few statements in the revised manuscript (Page 9, Line 1 from the bottom):

“ MAPbBr_3 generally possess a higher static distortion compared to the other two by around 10 meV, which should be due to the centrosymmetry breaking by MA cations. Furthermore, it is interesting to note without thermal distortion at low temperature, CsPbBr_3 SCs exhibits a static splitting of around 26 meV, implying the contribution of surface distortion is significant in causing Rashba splitting at low temperature.”

(3) A similar unspecified remark is mentioned in the abstract line 28 (and not mentioned at other places): indirect 'Urbach-like' states. What are those states?

Response: Previously, the term indirect 'Urbach-like' states was used to refer to the novel tail states in lead-halide perovskites sharing characteristics of both indirect band gap states and Urbach tail states. The conduction band minimum is momentum mismatched with respect to the valence band maximum (hence indirect), and they are largely from the lattice fluctuation (similar as the origin of Urbach tail states). To avoid confusion to the readers, we cautioned the use of this term. We now refer to these states as 'indirect tail states' instead of 'indirect Urbach-like' states for greater clarity. We have corrected all the terms and included an explanation in the manuscript.

(4) Seems that something is wrong of the color coding of Figure 2F. In the present case not much difference is found for the two emission peaks, while in the text they mention the there are huge changes.

Response: there is no color problem in Figure 2f. The huge difference of the PL kinetics occurs at low temperature (165 K). But the PL dynamics is almost identical at room temperature.

(5) In the orthorhombic phase of MAPbBr_3 a huge broad emission peak is observable above 600 nm, which is unfortunately cut off, which is probably related to radiative process 6 in Figure 4e.

Response: Yes, there is a broad emission around 600 nm for MAPbBr_3 SC. However, for CsPbBr_3 and FAPbBr_3 , we did not observe this broad emission. The emission also disappears after the temperature is elevated to beyond the phase transition temperature (~ 150 K). So the emission is most probably attributed to the interaction of the MA cation and the carriers, *i.e.*, the dipolar polaron emission, or carriers trapped by MA cations. The exact origin of this emission is still unknown, but we can generally rule out the relevance of this emission to the concept we discussed in this manuscript.

Reviewer #2 (Remarks to the Author):

The authors describe spectroscopic measurements of bromide-based lead halide perovskite single crystals with different organic cations (Cs, MA, FA) and study in detail the sub-bandgap states. They vary the temperature, excitation density and wavelength, and morphology, and find that the below bandgap feature present in PL and absorption is linked to the indirect transition generated via the Rashba effect.

Generally, the data is interesting and the interpretation in many places makes sense. However, the writing is very bad in some places, and the paper is oversold in others. Also, many findings are not as novel as the authors claim they are. Thus, while the paper should be published, and might be suitable for Nature Communications, major revisions are necessary, both in writing and interpretation.

My main concern is the claim of novelty. The data quite nicely shows the Rashba effect, but all the effects have been observed in similar systems with similar materials. E.g., as the authors acknowledge, the low-energy peak in the PL has been assigned to the indirect gap before (in MAPI, DOI: 10.1039/C6EE03474H), the low-temperature dependence has been studied before (in MAPI, doi:10.1038/nmat4765), the Rashba effect has been observed in bromide perovskites by ARPES (/doi.org/10.1103/PhysRevLett.117.126401) etc.; Many of the claims are not new, the main novelty of the paper in my view is the fact that it compares different cations, and single crystals to thin films. The authors should tone down the paper and work out exactly what is novel about their study. This would also mean that the big claims about the far reaching impact of their findings would need to be toned down.

Response: We thank the reviewer for the careful reading, useful comments and precious suggestions. We appreciate that the reviewer shares our enthusiasm of our findings and feels that the manuscript is suitable for *Nature Communications* after revisions. We do sincerely apologize for our excitement depicted in the overenthusiastic style of writing. We have since corrected it.

However, we respectfully disagree with the reviewer about the novelty of the work. Indeed, as highlighted by the reviewer, Rashba splitting effects have been experimentally observed in a few literature reports. However, there are still significant gaps in understanding the origin of the Rashba splitting effects. Only the MA system is studied in all of the following works: doi:10.1039/C6EE03474H, doi:10.1038/nmat4765 and doi.org/10.1103/PhysRevLett.117.126401 and the origin of the splitting effect is attributed to the centro-symmetric breaking by the non-spherical organic cation.

Here in our manuscript, we systematically investigated the splitting effect by varying the A-site cations, temperature and crystalline size. We observed that the A-site cations actually play only a small role in determining the splitting effect (~ 10 meV). Instead, thermal-induced polar fluctuation dominates the splitting at room temperature. This contrasts with all those previous reports. The novelty of our manuscript are: (1) We elucidated the origin of the Rashba splitting effects in great detail. (2) We discerned the contribution to the Rashba splitting by A-site cations, surface effect and thermal fluctuation. (3) We found the thermal fluctuation is the most important contribution in causing the Rashba splitting at room temperature for high quality crystals, rather than A-site cations or surface distortion. (4) This thermal fluctuation induced dynamical Rashba effect give rise to indirect tail states in lead-halide perovskites that are not present in other semiconductors. These states have profound effects on the carrier properties in lead halide perovskites such as low e-h recombination coefficient, long carrier lifetime, high ASE threshold, high exciton-dissociation efficiency *etc*, that satisfactorily explains the observed phenomena in the literature.

Furthermore, now we have provided new evidence of the thermal polar fluctuation and its intimate

relation with the dual emission characteristic by analyzing the zero-frequency Raman data which is a signature of the lead-halide polar fluctuation. We observed the congruence of the temperature dependence of the splitting effect and the lattice polar fluctuation. We acknowledge that the significance and novelty of the findings in our original manuscript may not have been adequately highlighted, which could have affected the reviewer's view. We admit in the previous version, the novelty of the findings in our manuscript was not highlighted, which may affect the reading and the judgment. We have since revised the manuscript and improved the writing style of the manuscript. We are confident that the quality and novelty of our revised manuscript will meet the stringent standards of *Nature Communications*.

Additional detailed comments that need addressing:

1) *The abstract and the beginning are written terribly, like a sales-piece. Phrases like “enormous potential for various impactful deep tech applications” have no place in a serious scientific article. There are many of such phrases and buzzwords, and also many spelling mistakes. Also terms like “giant SOC” suggest technical terms, but here giant is presumably just used as synonym for large.*

Response: We thank the reviewer for their precious comments on the writing. We have revised the manuscript accordingly:

“optoelectronic properties with enormous potential for various impactful Deep Tech applications” has been changed to “amazing photophysical properties with enormous potential for various optoelectronic applications.”

“...giant SOC...” has been changed to “...large SOC...”

2) *On p.3, when discussing the various differences between SCs and TFs, the authors should cite works for each of the examples.*

Reply: We thank the reviewer for their feedback. We have inserted the relevant citations at the appropriate places. The references are:

- 16 Fang, X. *et al.* Effect of excess PbBr₂ on photoluminescence spectra of CH₃NH₃PbBr₃ perovskite particles at room temperature. *Appl Phys Lett* **108**, doi:Artn 071109 (2016).
- 20 Fang, Y. J., Wei, H. T., Dong, Q. F. & Huang, J. S. Quantification of re-absorption and re-emission processes to determine photon recycling efficiency in perovskite single crystals. *Nat Commun* **8**, 14417 (2017).
- 23 Wang, T. *et al.* Indirect to direct bandgap transition in methylammonium lead halide perovskite. *Energ Environ Sci* **10**, 509-515, (2017).
- 26 Cho, H. C. *et al.* Overcoming the electroluminescence efficiency limitations of perovskite light-emitting diodes. *Science* **350**, 1222-1225, (2015).
- 27 D'Innocenzo, V., Kandada, A. R. S., De Bastiani, M., Gandini, M. & Petrozza, A. Tuning the Light Emission Properties by Band Gap Engineering in Hybrid Lead Halide Perovskite. *J Am Chem Soc* **136**, 17730-17733, (2014).
- 28 Milot, R. L., Eperon, G. E., Snaith, H. J., Johnston, M. B. & Herz, L. M. Temperature-Dependent Charge-Carrier Dynamics in CH₃NH₃PbI₃ Perovskite Thin Films. *Adv Funct Mater* **25**, 6218-6227, (2015).
- 32 Myagkota, S. V., Gloskovskii, A. V. & Voloshinovskii, A. S. Photo- and X-ray luminescence spectra of CsPbX₃ microcrystals dispersed in a PbX₂ (X = Cl, Br) matrix. *Opt. Spectrosc.* **88**,

538-541, (2000).

- 34 Wenger, B. *et al.* Consolidation of the optoelectronic properties of CH₃NH₃PbBr₃ perovskite single crystals. *Nat Commun* **8**, 590 (2017).

3) *In several places do the authors claim that the indirect bandgap would be beneficial for photovoltaics. This is only true if the devices are limited by the mobility of the charge carriers. As shown by Rau & Kircharz (DOI: 10.1021/acs.jpcclett.7b00236), this is not the case for perovskites, where a completely direct bandgap would lead to higher efficiency.*

Reply: We thank the reviewer for highlighting the nice theory paper by Rau & Kircharz, which we have unfortunately missed out. The reference provides a deep physical discussion about the efficiency change when introducing an indirect band gap slightly below the direct band gap. In the radiative limit, they concluded that there will not be any benefit with such an indirect gap. We have revised the manuscript by adding lines in Page 16, Line 3:

“This ensures long carrier diffusion lengths favorable for charge extraction in a solar cell with relatively low carrier mobility. On the other hand, for high and balanced carrier mobilities such as in optimized lead halide perovskites, it may not help to increase the ultimate efficiency in the radiative limit.⁵²”

4) *When discussing Figure 1d, the authors do not give a reason as to why they do not see the low-energy peak in the thin films. Presumably, the absorption is just too weak?*

Reply: We thank the reviewer for pointing this out. Indeed, for thin films with ~ 100 nm thickness, the absorption from the indirect transitions is insignificant as the indirect absorption coefficient is typically < 10³ cm⁻¹. It will require a thick film of several microns to get sufficient light absorption. For emission profiles, the low-energy PL intensity is highly dependent on the crystalline quality, as shown in Figure 2a. This is due to the carrier distribution and *e-h* recombination coefficient that is sensitive to the Rashba splitting, doping and other local environmental effects, *etc.* In general, the high energy peak (Peak 1) is direct but with less carrier occupation; while the low energy peak (Peak 2) is slightly indirect but with higher carrier occupation. Hence, in some cases, the two peak intensities can be comparable.

In view of the reviewer's comment, we have included a few lines in Page 5, Line 1 from the bottom:

“...indicating band edge emission in the TF. The Peak 2 is absent in TF due to its weak absorbing and emitting nature. The EQE...”

5) *For the same figure, why is the EQE shape so strange? i.e., why is there a peak right at the band edge?*

Response: The peak at around 560 nm in EQE near the direct band edge is evidence of the indirect tail states. The peak corresponds to the tail-state absorption by the Rashba splitting. Since it originates from an indirect transition, light can penetrate deep and excite the bulk of the crystal. Carriers generated can drift/diffuse to the respective electrodes and be collected. The distinct peak indicates (1) the transition is weak and (2) the carriers generated are delocalized. Above the direct band edge, the transition is direct. The photo-carriers are mainly generated near the front side of the crystal and therefore only one type of the carriers is predominantly collected. Hence, the difference in the EQE shape for these two cases of indirect

tail states and direct band edge transistions.

In response, we revised the manuscript accordingly in Page 5, Line 3:

“MAPbBr₃ SC based photo-detector shows high efficiency near Peak 2, indicating weak transition and delocalized nature of the excited species. The absorption is weak, so the light can easily penetrate deep into the crystal to reach the back side. Both types of carriers can therefore be collected at their respective electrodes. Another requirement for the efficient collection of carriers near Peak 2 is that they should be highly delocalized (free carriers) with long diffusion lengths.”

6) Far too little detail is given about the experimental details. Basic measures, e.g. how big are the single crystals, what is the difference in the two crystals shown in the SI for each material, what is the geometry of the photodetector etc. are omitted.

Response: We thank the reviewer for the comment. There is no difference in the two crystals in Figure S1. They are from the same batch. The single crystals used in our experiments have typical size of around 2-3 mm × 2-3mm × 0.5-1 mm. In our measurements, we did not find any obvious difference between the bigger and the smaller crystals. The photo-detector is made by placing a MAPbBr₃ SC (~ 0.5 mm) on Ga electrode and then evaporating a 28 nm thick Au electrode. The EQE is measured with 40nW/cm² light intensity and 4V bias. We have revised the manuscript accordingly in the experimental part.

7) One of the nicest findings of the paper is the difference in the dynamics of the two peaks. The indirect peak has slower dynamics at low T and identical dynamics at RT. Can the authors expand on the interpretation, for example calculate how much of the emission they expect to be in the direct/indirect peak at a certain T, form the energy difference and the phonon DOS.

Response: We thank the reviewer for his valuable comments to help us improve the manuscript. Conventionally, one would expect the hot carrier relaxation process assisted by phonons to be ultrafast (within tens to hundreds of femtoseconds; even for low temperature), which are several orders faster than the radiative recombination rate. It will be normal to see the lifetime is the same (like what we observed at room temperature 295 K) as the population exchange rate between the high-lying and low-lying states is rapid. Hence, it is very strange to see that the PL lifetimes can be different for Peak 1 and Peak 2 at 165 K and other low temperatures. We noticed a few publications proposed the so-called large polaron effect and phonon bottleneck effect that impedes hot carrier relaxation in MAPbBr₃ as in MAPbI₃ and FAPbI₃ (e.g., *Nat. Photonics*, 10, pages 53–59 (2016), *Nat. Comm.*, 6:8420 (2015), *Nat. Comm.* 8:14120 (2017)). This effect occurring on long timescales up to a few nanoseconds, may be one reason of the slow population exchange at low temperature. However, at current stage, the mechanisms are a matter of intense debate. Hence, we are unable to propose a model that could perfectly explain the observed phenomena that ranges from sub-ps to few ns. Further clarifications are needed.

8) From the ARPES measurements on MAPbBr3 one would suggest the Rashba splitting to reside mainly in the valence band (doi.org/10.1103/PhysRevLett.117.126401), yet from the DFT calculations here, and from the schematic figures, the authors suggest that the conduction band is more split. Can you explain that

discrepancy?

Response: In principle, both CBM and VBM can have Rashba splitting effects. The CBM is dominated by Pb p orbital, while the VBM is comprised of the anti-bonding states of Br p orbital and Pb s orbital. As Pb is larger than Br ($Z=82$ vs. $Z=35$), it is expected that the CBM have a much higher splitting effect than VBM. For example, in calculating the instantaneous band structure in Figure 3c at 300 K, we found the maximum Rashba splitting occurs in Γ -Y direction with a value 1.365 eV \AA at CBM. On the other hand, VBM shows negligible splitting with maximum value of only 0.22 eV \AA .

ARPES can only measure the VBM characteristics. It is reasonable to infer that the CBM should have a much higher splitting than the VBM if it could be measured. In response, we added a few lines in Page 8, Line 5:

“The Rashba band splitting at the VBM has been directly verified with angle-resolved photo-electron spectroscopy.¹⁴ The effect should be stronger at the CBM according to DFT calculations because the atomic number of Pb ($Z = 82$), which constitutes the conduction orbitals, is much larger than Br ($Z = 35$).”

9) In Figure 4 b & c the authors compare different single crystals, but never tell us what the difference is!

Response: We thank the reviewer for pointing out this potential ambiguity. The different crystals refer to three different MAPbBr₃ crystals made in the same batch. Even though the fabrication conditions are the same, the PL characteristics at different points can vary significantly at 77 K. The PL characteristics however can vary significantly at 77 K. This implies different surface distortion at different points or different samples may introduce different Rashba splitting effect at low temperature that causes the PL difference.

To avoid ambiguities, we revised the manuscript in Page 15:

“...splitting at low temperature (77 K). Note the SCs used in these measurements were prepared under the same conditions. Different surface distortions among different crystals and points may account for the different Rashba splitting effect and PL properties at low temperature. These results suggest the ease of lattice distortion and the associated change of the indirect tail states play a significant role in the emissive properties (PLQY, dual peaks, etc) of lead halide perovskites. The presence...”

10) Towards the end of the paper the authors speculate on the indirect gap to be beneficial for exciton dissociation. It is not clear to me how their interpretation makes sense. They claim that because the electron relaxes into the indirect CBM. How would that help to dissociate the exciton? Only if one changes the delocalization in real space would the binding energy be reduced, and I do not see how this could be done by thermalization into the CBM.

Response: Thanks the reviewer for pointing out this. After reconsideration, we would like to caution our original statement. However, the efficient exciton dissociation may be still relevant to the thermal structural fluctuation in the lead halide perovskite system. The inorganic lattice fluctuation is essential for determining the efficient charge separation in lead halide perovskites with or without organic cations (*J. Phys. Chem. C* 2017, 121, 26648–26654; *Nano Lett.* 2015, 15, 248–253; *Phys. Chem. Chem. Phys.*, 2015, 17, 9394). In response, we deleted the previous speculative part and revised the manuscript by adding the new discussion:

“We interpret that the photogenerated charges are rapidly separated due to electrostatic potential fluctuation coupled to the inorganic lattice dynamics. The organic cations which were believed to be critical in causing potential fluctuation and charge separations previously,^{66,67} may not be essential as H. Uratani and K. Yamashita demonstrated through MD simulations.⁶⁸”

Reviewer #3 (Remarks to the Author):

In this paper authors are trying to revised their interpretation of results based on few new experimental findings. Many of their argument fails to their own statements put up into abstract and introduction of paper. In general, in this paper authors are trying to put up a model based on co-existence of direct and indirect bandgap in perovskite semiconductor as an origin of observation of double emission peak. They argued in introduction that there are puzzling results and explained using various models and many time morphology was held responsible for these observations. Which I found is the case even for them where they comment that double emission peak observation is possible in their SCs due to good quality of crystal, which is backed up showing higher decay time as compare to their own previous results, However, there are reports which has also used similar technique to grow SCs and got lifetime higher or comparable to reported in this paper but still shows single emission peak in perovskite single crystals. Introduction should align with their results and discussion. I noted few things which needs attention from authors.

Response: We thank the reviewer for the comments. However, we respectfully disagree with his comments that our findings are not consistent with literature reports. Unfortunately, the reviewer has not provided any specific reference to this. Below are some data from the literature that reported single emission peak (Figure d,e,f), while others showed two peaks (Figure a,b,c). We wish to highlight that for those reports showing single peak, the single peak is in fact not symmetric. One needs to carefully examine the peak to notice the tail at the low-energy side (Figure d,e,f). These observations are in agreement with our findings that there is a weak low energy peak, whose intensity is relatively weak and subdued by the strong main peak at room temperature.

Figure R2. The emission profiles of MAPbBr₃ single crystals in the literature.

1. *Abstract is too general in the form of perspective instead of communication paper and fail to give the flavor on content of paper.*

Response: We thank the reviewer for pointing this out. We have modified the abstract accordingly.

2. *Whole PL needs to be fitted with Gaussian peaks to show whether they need 2 or 3 Gaussians to get physics out?*

Response: We thanks the reviewer for the raising this point. The whole PL spectra were fitted using Voigt (including Gaussian + Lorentz) functions. In response, we added one line in Page 5:

“...**Figure 1e** shows the temperature-dependent PL peak positions of CsPbBr₃, FAPbBr₃, and MAPbBr₃ SCs versus their TF counterparts **extracted with Voigt function**...”

3. *How does Peak1 and peak2 position changes individually with respect to temperature is missing? Though more suitable would be fit the Gaussians and then analyze.*

Response: The individual peak 1 and 2 position changes can be found in Figure 1e. The

temperature-dependent positions of Peak 1 and Peak 2 are extracted by fitting the PL spectra with Voigt functions. The Voigt function is more suitable than Gaussian function as homogeneous broadening and inhomogeneous broadening may both play a role in determining the linewidth of the PL spectra.

4. *Peak2 vs Peak1 splitting seems to be asymmetric, though both of them have lattice-dilation and electron-phonon interaction?*

Response: We thank the reviewer for the comment. The effects of lattice-dilation and electron-phonon interaction on the band gap of lead halide perovskites are complicated. Previous studies focused on the abnormal development of Peak 1 with temperature that is opposite to the Varshni relation (e.g., *Phys.Chem.Chem.Phys.* 2014, 16, 22476). They observed a blue-shift of the band gap (Peak 1) with increased temperature, *i.e.*, the lattice-dilation effect dominates in lead halide perovskites, in contrast to conventional inorganic semiconductors such as Si, GaAs. However, this anomalous behavior can now be explained based on our findings of the indirect tail states below the direct band gap, which is strongly dependent on electron-phonon interaction and red-shifts with elevated temperature. For example, Peak 2 of CsPbBr₃ between 77 K and 300 K (without any phase transition) can be well-fitted by the Varshni equation:

$$E_g = E_0 - \alpha T^2 / (T + \beta),$$

where $E_0=2.32 \pm 0.01$ eV, $\alpha=4.9 \pm 0.5 \times 10^{-4}$ eV/K, $\beta = 190 \pm 60$ K.

The energy difference between Peak 1 and Peak 2 is proportional to the phonon occupation number. That means, in lead halide perovskites, the fundamental gap (Peak 2) is also determined by both lattice-dilation and electron-phonon interaction effects. The electron-phonon interaction effect was not found previously because it results in indirect transitions with weak absorption coefficient that was not discerned. In view of the reviewer's question, we have added a few lines on Page 5, Line 13:

“However, Peak 2 exhibits red-shift with increasing temperature, which can be well-fitted by the Varshni equation for most inorganic semiconductors accounting for the electron-phonon coupling effect (Figure S3).”

And we added the figure below to the supporting information Figure S3.

Figure R3. The temperature-dependent Peak 2 fitted with the Varshni equation.

5. *How does their FWHM changes as a function of temperature for these Gaussians?*

Response: The FWHM for Peak 1 and Peak 2 follows the same trend. The peaks were fitted with Voigt function and the FWHM was defined as: $FWHM = \sqrt{wG^2 + wL^2}$, where wG and wL are the Gaussian and Lorentz linewidths, respectively. Figure R4 a shows the FWHM of a typical CsPbBr₃ SC, in which an LO phonon energy of around 20 meV can be globally extracted. For SCs, the Peak 2 FWHM may also be changed by reabsorption effect, so we turned to the FWHM of the films, i.e., the MAPbBr₃ film (1 M concentration) with obvious double peaks, which still shows very similar trend with temperature. This indicates the two peaks' broadening shares the same scattering mechanism dominated by Pb-Br LO phonon scattering at elevated temperatures (*Nat. Comm.* 7:11755 (2016)). In response, we have revised the manuscript accordingly in Page 10, Line 5:

“...The fitted values of E_{ph} are 12-18 meV, which corresponds well with the broadening of the PL peaks (Figure S7), and is attributed to the Pb-Br LO phonon modes (Pb-Br-Pb bending and Pb-Br stretchings).^{29,47,,}

And inserted Figure R4 as Figure S7 in the SI.

Figure R4, Temperature-dependent FWHMs of (a) CsPbBr₃ SC and (b) MAPbBr₃ TF.

6. *Instead of showing the TRPL for whole by separating faster and slower components, authors should compare TRPL due to each single Gaussian peak by deconvolution of the PL.*

Response: We thank the reviewer for the useful comment. For MAPbBr₃ thin film at room temperature, the population exchange between the two bands are much faster than the recombination, so the dynamics are identical between the two bands. We cannot separate them using any deconvolution method. For the dynamics at low temperature, due to slower population exchange (which is possibly related to slow hot carrier cooling in perovskites), we can separate the two band dynamics using non-negative matrix factorization method to decompose the two band dynamics as shown below:

Figure R5. The deconvolved TRPL profiles and dynamics of MAPbBr₃ thin film at 165 K using non-negative matrix factorization (NMF) method.

In response, we have used Figure R5 as Figure 2e-f and revised the main text accordingly in Page 9: “We employed the non-negative matrix factorization (NMF) method to deconvolute the TRPL data at 165 K. The deconvoluted effective lifetimes are around 110 ± 10 ns and 1540 ± 50 ns for Peak 1 and 2, corresponding to the recombination through the direct transitions and indirect transitions, respectively. At room temperature, both peaks show identical bi-exponential decay, indicating fast population exchange between the upper direct bands and lower indirect bands by phonon scattering.”

7. Fig 3a, ratio of peak1 to peak2 for Cs is in contrast to other two, where according to their model indirect peak has more PL than FA and MA based SCs, why?

Response: At this low temperature, the relative peak intensities are now sensitive to the excitation density. As shown in Figure R6 below, the low-energy PL peak relative intensity increases with excitation density. This is probably due to the slow population exchange at the two bands at low temperature as discussed above.

Figure R6. Intensity-dependent PL profiles of CsPbBr₃ at 45 K.

With our newer results, we have removed figure 3a from the original manuscript and relegated it to the supporting info. However, it does not discount our claims.

8. Fig. 3b shows the slope for split energy vs T is independent of molecular cation, which is very surprising and cannot be explained using model described by authors where lattice dynamics of these three system has to be different. However, reabsorption model can explain it in good way considering equivalent parameters in terms of absorption coefficient, thickness of samples, lattice dilation properties and possible scattering factor.

Response: We thank the reviewer for raising the important question. However, we respectfully disagree with the view about the reabsorption model. Our reasons are as follows:

The band structures near the band edges are dominated by PbX₆ octahedra, *i.e.*, the CBM is comprised of Pb p orbital and the VBM is the anti-bonding state of Pb s and X p orbitals. The energy levels of MA, FA, and Cs is far away from the band edges. Hence, it would be expected that the changing of the A-site cations will not alter much of the band edge properties and their dependence on temperature. The temperature-dependent trends of the ΔE in lead-bromide perovskites reflect the thermal deformation of the PbBr₆ octahedra, which is directly related to the occupation number of Pb-Br phonons.

One may expect the A-site cation motion can efficiently couple to PbBr₆ octahedra deformation, and may work indirectly on the observed temperature-dependence of the ΔE . However, we did not observe any

frequencies of Cs, MA or FA phonons coupled to Pb-Br modes to change the ΔE in our observation. This is consistent to that only the Pb-Br LO phonon modes via Fröhlich coupling contributes to the electron-phonon scattering and PL broadening in 3D perovskites (*Nat. Comm.*, 7:11755, (2016)), rather than an mixing of A-site cation phonon modes. A few recent published theoretical works also support our assumptions here. For example, Uratani and Yamashita compared the molecular dynamics of CsPbI₃, FAPbI₃ and MAPbI₃, in which they observed similar lattice fluctuation and efficient charge separation (*J. Phys. Chem. C* 2017, 121, 26648–26654). McKechnie et al predicted similar spin splitting effect in CsPbI₃ and MAPbI₃ (arXiv:1711.00533v1).

To verify this, we also simulated the root-mean-square atomic displacements of MAPbBr₃ and CsPbBr₃ in cubic phase at 500 K as shown in Figure R1 above. The RMSDs at 500 K for MAPbBr₃ atoms are (MA: 0.95 Å, Pb: 0.51 Å, Br: 0.86 Å), and for CsPbBr₃ atoms are (Cs: 0.74 Å, Pb: 0.54 Å, Br: 0.65 Å). From this, we believe the change of A-site cation from Cs to MA may not significantly change the PbBr₆ octahedra dynamics.

The reabsorption factor can be well excluded as we discussed the high quality thin film part in the main text. For thin film on the order of 200 nm, it is impossible to yield any significant reabsorption, while we can still get Peak 2.

In response, we added discussion in Page 11, Line 9 from the bottom.

“Although the inorganic and organic A-site cations have different vibrational characteristics, our findings show that they do not play an important role in determining the temperature-dependent Rashba splitting. Their phonon frequencies are not coupled to the Pb-Br frequencies to collectively influence the temperature trend of the splitting. To further validate this, we compared the root-mean-square displacements: $\text{RMSD} = \sqrt{\frac{1}{N} \sum_{i=1}^N (r_i - r_{0i})^2}$ where N is the number of the equivalent atoms, $r_i - r_{0i}$ is the displacement of the i_{th} atom from its equilibrium position r_{0i} . The time-dependent trends of each atoms in MAPbBr₃ and CsPbBr₃ cubic supercells are shown **Figure 3e** within 15 ps. The RMSDs at 500 K for MAPbBr₃ atoms are (MA: 0.95 Å, Pb: 0.51 Å, Br: 0.86 Å), and for CsPbBr₃ atoms are (Cs: 0.74 Å, Pb: 0.54 Å, Br: 0.65 Å). This indicates the change of A-site cations (*i.e.*, MA, FA and Cs) here does not affect the Pb-Br displacements much. It would then be reasonable to conclude that any phase changes that are directly linked to the motional freedom of A-site cations should also not play a role in the splitting effect. Note the...”

8. *This is known that there is good amount of scattering in these crystals and temperature dependent lattice expansion, both of these factors can influence their calculation of reabsorption model, as this is one of the strong model to explain the dual emission peak in literature as referred by authors as well.*

Response: Reabsorption effect is an important factor that may contribute to the optical behavior of the thick single crystal. However, it is not sufficient to explain the observed phenomena in thin films as discussed in **Figure 2** in the manuscript. Similar double PL peaks were also reported for MAPbI₃ thin films before (*Energy Environ. Sci.*, 2017, 10, 509).

The thin films look very smooth by eye and from the AFM images (Figure R7) The RMS are 4.8 nm, 4.8 nm, 7.7 nm and 6.5 nm for thin films with 0.25 M, 0.5 M, 0.75 M, and 1 M precursor concentration, respectively. It is unlikely the scattering effect play an important role. The absorption

coefficient measured is also in good agreement with other reports using ellipsometry (*Nat. Comm.*, 6:7961 (2015)). We fully considered the reabsorption effect as the possible reason of the double peaks; however, it fails to explain the observed phenomena satisfactorily.

In response, we have added a few lines in Page 7, Line 13:

“- free electron-hole recombination. Furthermore, the optical scattering related artefacts are not likely to occur in the TFs as they look very smooth from AFM images with typical RMS less than 8 nm in $10\ \mu\text{m} \times 10\ \mu\text{m}$ region (Figure S5). These observations hint at a common origin as those in SCs.”

And insert Figure R7 as Figure S5 in the SI.

Figure R7. AFM images MAPbBr₃ films prepared with varied precursor concentration.

9. Authors claim about Rashba splitting is indirect, where unusual polaronic effects and defect tolerant perovskite semiconductors can make these interpretations illusive. A more direct approach needed for this big claim. (doi:10.1038/nphys675)

Response: Our claim about the Rashba splitting is strongly supported by many theoretical and experimental works. Direct approaches such as magneto-optical measurements have been applied to observe the PL fine structures of CsPbBr₃ nanostructures due to the Rashba effect at extremely low temperature (*Nano Lett.*, 2017, 17, 5020–5026; *Nano Lett.*, 2017, 17, 2895–2901). Another powerful technique angle-resolved photoemission spectroscopy (ARPES), has also been used to observe clear signature of the valence band splitting in MAPbBr₃ single crystal due to the Rashba effect (*Phys. Rev. Lett.*, 117, 126401 (2016)).

Furthermore, we have performed temperature-dependent Raman scattering measurements from

which we can observe clear temperature-dependent trend of the zero-frequency Raman signifying the polar fluctuation of lead halide perovskites. We observed direct correlation between the dual emission (splitting effect) and the thermal polar fluctuation as shown below: The polar fluctuation becomes stronger as the increase of temperature. The zero-frequency mode intensity undergoes a similar increase fashion with that of the energy splitting.

Figure R8. (a) Temperature-dependent low-frequency Raman spectra of CsPbBr₃ SC. (c) Temperature-dependent relative intensity of the zero-frequency mode.

In lead halide perovskites, the (large) polaronic effects are often mentioned and are responsible for their defect tolerance and screened scattering. (e.g., *J. Phys. Chem. Lett.* 2015, 6, 4758–4761, *Sci. Adv.* 2017;3: e1701217) In this manuscript, we do not attempt to prove whether the photo-excitation in lead halide perovskites is free carrier or polaron (carrier dressed with lattice deformation). In the lead halide perovskites whose lattices are apt to deform, it is very probable that the photo-excited carriers can form polarons. However, we note that whether free carrier or large polaron in lead halide perovskites does not contradict the band splitting effect. If the photo-excitations are treated as polarons, the polaron bands are still possible to be split in the presence of the Rashba effect.

Lastly, one may argue there is the coexistence of free carrier and polaron, and that the high-energy peak is due to free carrier whilst the low-energy peak is from polaron. However, our optical signatures can generally rule out this case. For example, the weak tail state absorption edge is consistent with the low-energy PL peak. We know from polaron theory that it is the quasi-particle of excited carriers with the perturbed lattice, so that it could not have ground-state absorption signature near the band gap. It implies the low-energy PL peak could not be due to polaron emission, either. In addition, if the low-energy peak is understood as from polaron, as temperature increases, the polaron should be expected to ionize to a free electron and renormalized phonons (*Phys. Rev. B*, (12), 5472, (1975)). That is, the energy difference between the free carrier and polaron should reduce with temperature.

In response, we have inserted the Raman spectra related discussion in the main text and SI.

10. *Determining the Urbach Tail in SCs is very risky considering (i) the scattering components (ii) limitation due to cryostat (iii) thickness of highly absorbing samples, one can always miss actual absorption edge and will probe only deep states.*

Response: We agree with the reviewer about the difficulty of determining the Urbach tail in SCs. Diffuse reflectance is still the main technique to measure the absorption tail as detailed in many previous literature reports, *e.g. J. Appl. Phys.* 78(9), 5609 (1995). Although we cannot quantify the absorption coefficient using the diffuse reflectance spectroscopy, it provides information where the fundamental absorption edge is. The factors the reviewer mentioned can indeed affect the measured tail absorption, but we can generally exclude them through careful measurements. For example, (i) the scattering that is not relevant to the band gap usually behaves as a flat background, which can be simply removed. (ii) The cryostat effect can be excluded as we compared the diffuse reflectance spectra measured with and without the cryostat and we did not observe any difference. (iii) Since we used diffuse reflectance spectroscopy to measure the absorption edge in SC, there is inevitable weak absorption from sub-gap defect states. However, we observed the slope of the absorption edge shows a temperature-dependence, which is a typical signature of thermal fluctuation induced tail states (Figure R9). Hence, it is reasonable to conclude that we were measuring the tail state absorption.

Figure R9. (a) Temperature-dependence of the tail absorption slope for MAPbBr₃ SC. (b) A linear relation of $\ln(\alpha)$ on energy can be obtained.

12. Reference for inverted temperature crystallization needs to be relooked.

Reply: We thank the reviewer for pointing out the error. We have updated the references now:

“FAPbBr₃ and MAPbBr₃ single crystals were prepared using the reported inverted temperature crystallization (ITC) method.^{57,71}”

Reference 57 and 71 are:

- 57 Saidaminov, M. I. *et al.* High-quality bulk hybrid perovskite single crystals within minutes by inverse temperature crystallization. *Nat Commun* **6**, 7586 (2015).
- 71 Zhumekenov, A. A. *et al.* Formamidinium Lead Halide Perovskite Crystals with Unprecedented Long Carrier Dynamics and Diffusion Length. *Acs Energy Lett* **1**, 32-37,(2016).

13. Page 15, line 12 english error.

Response: thanks the reviewer for pointing out the typo. We have deleted the excess “and” in the revised manuscript.

“The temperature was and cooled down using liquid nitrogen” has been changed to “The temperature was

cooled down using liquid nitrogen.”

14. Diffuse reflectance as a function of temperature using Nikon microscope in dark field could not really explain how this measurement was performed, more details needed.

Response: We thank the reviewer for pointing this out. The temperature-dependent diffuse reflectance was measured using a Nikon microscope equipped with a liquid-nitrogen cooled cryostat (Janis). The white light from a tungsten bulb was directed onto the single crystal with 10× objective and the diffused light was collected with the dark field mode. The incident light intensity was calibrated by measuring the reflected light intensity from a quartz substrate and calculated with the as-known quartz refractive indices. Later the diffused light from the single crystals was collected and divided by the incident intensity to get the diffuse reflectance spectra.

We have added the details above into the experimental part of the revised manuscript now.

15. Detector EQE biasing conditions vs illumination direction and applied bias information is missing.

Response: The photo-detector EQE was measured with 40nW/cm² light intensity and 4V bias. We have added it in the revised manuscript now.

Considering above comments, I do not feel that this work full-fill the requirement of Nature Communication standard and should be published elsewhere after justifying the above comments.

Reviewer #4 (Remarks to the Author):

In NCOMMS-18-05870 the authors report an experimental, optical spectroscopic study on halide perovskites where the interplay between thermal atomic displacements and electronic properties is investigated.

By means of temperature dependent photoluminescence and reflection measurements, the authors reach the conclusion that strong spin-orbit coupling combined with symmetry breaking due to thermal fluctuations cause tail states in the conduction band (Due to a Rashba effect). These tail states are expressed as a dual peak in the photoluminescence spectra.

The study is comprehensive and has merit. The findings are important and deserve publication. Yet, there are several issues that must be resolved prior to publication:

Response: we sincerely thank the reviewer for the meticulous reviewing and positive evaluation of our work. We are delighted that the reviewer shares our enthusiasm about this work for *Nature Communications*.

1. The authors claim that the dual peaks in the photoluminescence are intrinsic to the crystal. Though they provide some evidence for their claim, there are evidence to the contrary. In DOI:10.1021/acsnano.6b02734. Tilchin et al report the temperature dependent PL and absorption of MAPbBr3 single crystals. In their data, the low energy peak is considerably less pronounced than here (compared to figure 3A at 45 K). The aforementioned paper is not cited by the authors (I am sure it was not ignored but missed). I am also aware of other unreported data (presented in conferences) where the low energy peak is even less pronounced. I wonder how this fact coincides with the authors interpretation. This point must be addressed in the text.

Response: We thank the reviewer for the comment. Unfortunately, we were not aware of the paper published by Tilchin *et al.* in *ACS Nano*. We did notice a few literature works showing less pronounced low-energy peak compared to the high-energy peak (Figure R2 above). It implies the peak intensity is highly sensitive to the local physical properties of the perovskites. These physical properties such as disorder/defects, lattice distortion and doping can determine the band structure and the carrier distribution at the direct band edge in the presence of the indirect tail states. The electron-hole recombination coefficient at the direct band edge of an indirect bandgap material can be expressed as⁴⁹:

$$B_1 = \frac{(2\pi)^{1/2} h e^2 n}{M_c c^3 m_0^{5/2} (k_B T)^{3/2}} \left(\frac{m_0}{m_e + m_h} \right)^{3/2} \left(1 + \frac{m_0}{m_e} + \frac{m_0}{m_h} \right) E_{gd}^2 \exp\left(\frac{E_{gi} - E_{gd}}{k_B T} \right)$$

where E_{gi} and E_{gd} are the indirect and direct bandgap, respectively, m_e and m_h are the effective masses of carriers in the conduction band and valence band, respectively. m_0 is the electron mass, h is Planck constant, k_B is Boltzmann constant, T is the temperature, e is the elementary charge, c is the light speed in vacuum, n is the refractive index and M_c is the number of equivalent minima in the conduction band. The calculated B_1 is sensitive to the exponential term $\exp((E_{gi} - E_{gd})/k_B T)$ determined by the direct-indirect bandgap difference.

Hence, the sensitivity of the high-energy/low-energy PL intensity is directly related to the direct-indirect bandgap difference, which cannot be satisfactorily explained with other theories. There is no contradiction to our claims.

In response, we have cited the paper mentioned and revised the manuscript in the main text and SI accordingly:

“The appearance of the dual emission would degrade the light emitting color purity. A large fraction of the carrier population is dispersed in the indirect tail with relatively lower transition probability. This points to the difficulty of obtaining highly-efficient light emission and lasing, at least in the high-quality SCs or large grain PCs. However, the soft nature of Pb-halide bonds makes the perovskite lattice apt to deformation by surface, disorder/defects *etc.*,³⁹ which disrupts the long-range ordering and changes the direct and indirect characteristics of lead halide perovskites. Likewise, the radiative recombination rate can be significantly altered with varied defect doping, electron-phonon coupling, or quantum confinement. These factors can complicate the emission properties that lead to non-prominent or prominent indirect/direct emissions.⁵⁵⁻⁵⁷ If we treat the indirect tail as an indirect band gap semiconductor with its average band gap position located at Peak 2, the electron-hole recombination coefficient at the direct band edge of an indirect bandgap materials can be expressed as⁵⁸:

$$B_1 = \frac{(2\pi)^{1/2} h e^2 n}{M_c c^3 m_0^{5/2} (k_B T)^{3/2}} \left(\frac{m_0}{m_e + m_h} \right)^{3/2} \left(1 + \frac{m_0}{m_e} + \frac{m_0}{m_h} \right) E_{gd}^2 \exp\left(\frac{E_{gi} - E_{gd}}{k_B T} \right) \quad (1)$$

E_{gi} (E_{gd}) are the indirect (direct) bandgaps. m_e (m_h) are the electron (hole) effective masses. m_0 is the electron mass, h is Planck constant, k_B is Boltzmann constant, T is the temperature, e is the elementary charge, c is the light speed in vacuum, n is the refractive index and M_c is the number of equivalent minima in the conduction band. The calculated B_1 is sensitive to the exponential term $\exp((E_{gi} - E_{gd})/k_B T)$ determined by the direct-indirect bandgap difference, and therefore can be significantly changed with different surface distortion and defect density. As shown in Figure 4a, we observed a striking difference of the effective e-h recombination coefficients between polycrystalline TFs and SCs (details of the fitting can be seen in Figure S11 and Supplementary Note 4). At room temperature, the effective e-h recombination coefficient of FAPbBr3 TFs approaches $10^{-8} \text{ cm}^3 \text{ s}^{-1}$ which is three orders higher than that of the bulk of the SCs. Similarly, the...”

2. I am not sure why the authors suggest that the Rashba splitting (that, the best of my understanding, is responsible for the low energy peak) occurs exclusively in the conduction band. To the best of my experimentally measured (DOI:10.1103/PhysRevLett.117.126401) for the valance band.

Response: We thank the reviewer for the valuable comment. As discussed above in response to the 8th

question by reviewer 2, both CBM and VBM in principle can have Rashba splitting effects. The CBM is dominated by Pb p orbital, while the VBM is comprised of the anti-bonding states of Br p orbital and Pb s orbital. As Pb is larger than Br, it is expected that the CBM have a much higher splitting effect than VBM. For example, in calculating the instantaneous band structure in Figure 3c at 300 K, we found the maximum Rashba splitting occurs in Γ -Y direction with a value 1.365 eV \AA at CBM. On the other hand, the VBM shows smaller splitting with maximum value of only 0.22 eV \AA , only 20% of that of the CBM.

As we know, ARPES can only measure the valence band properties. It would be expected the CBM should have a larger splitting effect than the VBM if it can be measured.

To avoid ambiguities, we made a few comments in the revised manuscript.

In Page 9, Line 5, we added:

“The Rashba band splitting at the VBM has been directly verified with angle-resolved photo-electron spectroscopy.¹⁴ The effect should be stronger at the CBM according to DFT calculations because the atomic number of Pb ($Z = 82$), which constitutes the conduction orbitals, is much larger than Br ($Z = 35$).”

3. Moreover, the Rashba splitting should be symmetrical around the gamma point (unlike what is depicted in figure 3d and 4e) and as the authors point out, creates an indirect band gap. On this point two issues are not clear to me:

Response: We thanks the reviewer for the valuable comment and pointing out the potential ambiguity in our schematic. For lead halide perovskites with organic cations, the splitting may be asymmetric due to different interaction of the organic cation and the octahedra at different directions (as seen the band structure below for MAPbI₃). For more generality, we have corrected this in the revised manuscript by aligning the CBM and VBM around the R point for a typical Rashba splitting effect.

Rashba effect lifts the spin-degeneracy and creates Fermi surface consisting of two concentric circles with opposite chiral spin textures. In lead halide perovskites, both conduction band and valence band have Rashba effect. However, since Pb is heavier than Br, it is expected the SOC effect is more severe at the CBM which is comprised of Pb p orbital than the VBM which is comprised of Br p orbital and Pb s orbital. The difference of the Rashba effect can lead to a slight momentum mismatch of the CBM with respect to the VBM. Therefore, the consequence is spin-split indirect band gap is formed. This was illustrated in APL Materials **4**, 091501 (2016); doi: 10.1063/1.4955028 and as shown in Figure R10. In this calculation paper, the splitting effect is more severe at R->M direction compared to R-> Γ direction, which leads to an asymmetric CBM that resembles our earlier schematic.

Figure R10. Electronic band structure of $\text{CH}_3\text{NH}_3\text{PbI}_3$. From APL Materials 4, 091501 (2016).

In response, we changed the schematic of Figure 3d to:

a. How come the photoluminescence from the indirect gap is so efficient that it has the same order of magnitude of the direct band gap photoluminescence?

Response: We thank the reviewer for the interesting comment. The indirect PL intensity can reach the same order of magnitude of the direct gap PL due to a few reasons:

- (a) More carrier population at the low-lying indirect bandgap than at the high-lying direct bandgap. This is due to Fermi-Dirac distribution: $f(E) = 1 / (1 + \exp((E - E_F) / k_B T))$. The population can vary by orders which significantly enhance the intensity ratio of the indirect emission vs. direct emission.
- (b) Small momentum mismatch for the indirect emission. The CBM and VBM have only a minor momentum difference in the presence of the Rashba splitting effect. The momentum mismatch is much smaller compared to traditional indirect semiconductor such as Si. Hence, it would be expected the indirect transition can easily occur with phonons compensating the momentum mismatch.
- (c) Efficient electron-phonon coupling. There are many reports on the electron-phonon coupling in lead halide perovskites. The polar nature of the perovskites may lead to efficient indirect emission.

In response, we have added in the part “Implications for light emission applications.” of the revised manuscript:

“It is reasonable that the low-lying indirect emission peak can be as prominent as the direct one when more carriers reside at the indirect tail than in the direct band, and the phonon-mediated momentum mismatch is relatively small, i.e. relatively strong Fröhlich coupling may also assist the crystal momentum transfer in an indirect emission process.^{29,54}”

b. If low energy photoluminescence does indeed result from thermally induced Rashba splitting, shouldn't one expect an overlay of a distribution of split bands - meaning effectively broadened bands eventually - instead of a clear discrete split band?

Response: We thank the reviewer for this interesting question. There are three factors that determine the photoluminescence intensity. (1) The probability to find an electron at certain energy. The lower energy, the higher electron occupation probability. (2) The density of states at certain energy. Thermal fluctuation tends to create a distribution of density of states below the band gap. Hence, the lower energy, the less density of states. (3) The momentum matching conditions between the conduction and valence bands. Hence, the larger the Rashba splitting, the less probable the momentum matching can be reached.

Thermal-induced Rashba splitting creates a distribution of split bands. For the split bands with large thermal polar distortion, the electron occupation probability is high, but the density of states and momentum matching is reduced, and vice versa (Figure Rx). Therefore, a maximum transition can occur near the equilibrium position, creating a peak-like PL structure below the direct emission peak.

Figure R11. (a) The band structure with small lattice polar distortion and Rashba splitting. (b) The band structure with large lattice polar distortion and Rashba splitting.

In response, we have added in the part “Implications for light emission applications.” of the revised manuscript:

“The presence of thermal polar fluctuations and their associated indirect tail states can lead to the dual emission phenomena that were not observed in conventional polar semiconductors with an Urbach tail. Strictly speaking, the second emission peak at elevated temperature with thermal fluctuation should correspond to the energy position when the product of the density of the carriers and the transition probability reaches its maximum.”

Reviewers' comments:

Reviewer #1 (Remarks to the Author):

The manuscript has improved very much in clarity and reads very well. Although there are other works reporting parts of their results to completeness of this work including a sound description of the model justifies publication in nat com in my opinion.

Reviewer #2 (Remarks to the Author):

The authors have addressed all my remarks sufficiently, and have done a thorough job also addressing the extensive remarks of the other reviewers. The writing and scope has also improved significantly. The manuscript should be published in its current form in Nat Commun.

Reviewer #4 (Remarks to the Author):

Due to lack of time I read carefully only the authors response to my questions and the changes made in the ms.

I am not satisfied with the response and cannot recommend publishing the paper in its present form.

The main problem is that the authors' main claim is that Rashba splitting gives rise to the optical properties of the hybrid perovskites. I see no proof of that in the data. Their reply to my questions is a double-edged sword. On one hand, they claim that Rashba splitting is significant, on the other, they claim that it is very sensitive to surface and bulk defects and that the energy difference between the direct and indirect band gap is so small that the photoluminescence from the indirect band gap is as bright as the direct gap. That does not make sense in the same way that homoeopathy treatment does not make sense.

The authors did a good job of measuring the optical properties of quite a few crystals. I also tend to agree with their claim that local polar fluctuations play a key role in determining the electronic properties of these materials. However, if the authors wish to study the Rashba splitting they should measure it directly. I think that any measurement of an electronic property under magnetic field (e.g. DOI: 10.1021/acs.nanolett.7b02248 - this comes from the same group that wrote the paper the authors missed last time) can provide useful information.

Reviewer #5 (Remarks to the Author):

While this manuscript contains some interesting findings, it cannot be published as the main conclusions are (partially) based on a misinterpretation of spectroscopic measurements.

The authors report a difference in band gap for MAPbBr single crystals and thin films. However, a recent paper in Nature Communications showed very clear evidence that the optical band gap for MAPbBr in single crystals is identical to the band gap in thin films:

<https://www.nature.com/articles/s41467-017-00567-8>

This is in direct contradiction to the results in Fig. 1d. In the manuscript at hand, the problem arises most probably from using diffuse reflectance measurements to determine the band gap of the single crystals. Diffuse reflectance measurements depend on parameters such as grain diameter, surface

roughness, and refractive index. Deriving the band gap of a material from diffuse reflectance measurements is therefore likely to be inaccurate, see e.g.:
<http://www.sciencedirect.com/science/article/pii/S0927024807001948>

I note that several groups in the perovskite community have used diffuse reflectance spectroscopy for (doped) perovskite crystals and derived too small band gaps. In general, I want to point out the importance of carefully executed and properly interpreted optical spectroscopy. To obtain the optical properties of single crystal samples, it is advisable to use a specular reflection technique (such as ellipsometry). For thin films transmittance is often a good choice. Diffuse reflectance should only be used for very fine powders and relatively weak absorption.

With respect to the PL measurements, I am not convinced that the apparent peak shift is not due to self-absorption. A clear hint is the reported shift with increased precursor concentration (and hence higher film thickness).

Reviewer #1 (Remarks to the Author):

“The manuscript has improved very much in clarity and reads very well. Although there are other works reporting parts of their results to completeness of this work including a sound description of the model justifies publication in nat com in my opinion.”

Response: We thank the reviewer for all his/her valuable suggestions to improve our manuscript and recommendation for publication.

Reviewer #2 (Remarks to the Author):

“The authors have addressed all my remarks sufficiently, and have done a thorough job also addressing the extensive remarks of the other reviewers. The writing and scope has also improved significantly. The manuscript should be published in its current form in Nat Commun.”

Response: We thank the reviewer for his/her favorable comment that our manuscript is suitable to be published in Nat. Comm. We greatly appreciate the excellent suggestions the reviewer proposed to strengthen our work.

Reviewer #4 (Remarks to the Author):

“Due to lack of time I read carefully only the authors response to my questions and the changes made in the ms.

I am not satisfied with the response and cannot recommend publishing the paper in its present form.

The main problem is that the authors' main claim is that Rashba splitting gives rise to the optical properties of the hybrid perovskites. I see no proof of that in the data. Their reply to my questions is a double-edged sword. On one hand, they claim that Rashba splitting is significant, on the other, they claim that it is very sensitive to surface and bulk defects and that the energy difference between the direct and indirect band gap is so small that the photoluminescence from the indirect band gap is as bright as the direct gap. That does not make sense in the same way that homoeopathy treatment does not make sense.

The authors did a good job of measuring the optical properties of quite a few crystals. I also tend to agree with their claim that local polar fluctuations play a key role in determining the electronic properties of these materials. However, if the authors wish to study the Rashba splitting they should measure it directly. I think that any measurement of an electronic property under magnetic field (e.g. DOI: 10.1021/acs.nanolett.7b02248 - this comes from the same group that wrote the paper the authors missed last time) can provide useful information.”

Response: We thank the reviewer for his valuable comments. The presence of the Rashba splitting in lead halide perovskites have already been proven in many literature through various complimentary

techniques, to name a few, angle-resolved photoemission spectroscopy, circular photogalvanic effect, magneto-PL, etc.

We agree the experiments under magnetic field can be an important and straight-forward technique to prove the Rashba effect, as those done in DOI: 10.1021/acs.nanolett.7b02248. However, we note that such splitting by magnetic field is insignificant even with a 8T strong field – i.e., a splitting of only ~ 1.2 meV is observed. Therefore, it will be impractical to discern any change of the PL for our large-scale single crystals, which have typical PL full-width half maximum (FWHM) on the order of tens of meV.

An alternative approach is to use circular-polarized laser pulses to excite the single crystals and detect their spin-related properties. The Rashba effect lifts the degeneracy of the band edges (here mainly CBM for APbBr₃), forming two split spin valleys (Figure R1). The spin-flipping could thus be slowed down for carriers in the valleys due to a barrier formed between the two valleys with opposite spin. A circular-polarized (left-circular σ_+ , right-circular σ_-) excitation near the band edge will selectively excite one of the valleys. The ensuing PL will retain the helicity of the excitation depending on the timescales of the spin-flipping and recombination/transport.

Figure R1. Schematics of the Rashba split bands and selective excitation with circularly-polarized light.

Herein, for our new experiments, we excite the lead halide perovskite single crystals with circularly polarized laser pulses and detect the circularly polarized PL. There is an obvious difference in the intensity of the PL under different optical pumping helicity near the band edge, indicating the presence of two valleys that can be selectively excited. Typical right circularly polarized (σ_-) PL spectra of MAPbBr₃ SC following optical excitation with left (σ_+) and right (σ_-) circularly-polarized laser pulses at with 532 nm, 473 nm, and 400 nm lasers at 77 K are shown in Figure R2. With 532 nm pumping, the PL helicity is most prominent, which becomes reduced with 473 nm excitation and is negligible with 400 nm excitation. These results indicate that when the photocarriers are generated near the band edge, the carrier spin is preserved due to the potential barrier formed between the two spin valleys. Without the Rashba effect and the formation of two spin valleys, the strong SOC in lead halide perovskites will lead to an ultrafast spin-flipping and negligible PL helicity. Similar results can be obtained for CsPbBr₃ SC (Figure R2 d-f). Indeed, similar experiments were previously performed for 2D transition metal chalcogenides (DOI: 10.1038/NNANO.2012.96; DOI: 10.1038/nano.2012.95). We also note the similar experimental results were reported by Niesner *et al* in a recent online paper (DOI: 10.1073/pnas.1805422115), in which they measured the

spin-current of MAPbI₃ SC that follows the optical pumping helicity, the so-called circular photogalvanic effect. Therefore, we are very confident that our results provide strong evidence of the formation of spin-valleys and the Rashba effect in lead halide perovskites.

Figure R2. Right circularly polarized PL for (a-c) MAPbBr₃ SC and (d-f) CsPbBr₃ SC upon (a, d) 532 nm (b, e) 473 nm and (c, f) 400 nm optical excitation with right σ (blue) or left σ (red) circularly polarized laser pulses.

In response, we have added the following contents in the main text (Started from Page 9, Paragraph 2) and inserted Figure R2 d-f into the supporting information (Figure S7):

We further confirmed the formation of the split spin valleys through demonstrating the PL helicity that depends on the excitation light helicity at cryogenic temperature. The experimental schematic is shown in **Figure 3a**. If there are spin-split bands due to the Rashba effect, we can selectively excite these bands using circular-polarized optical pumping. To exclude any instrumental polarization-dependent response, we kept the same detection polarization and only varied the incident polarization only. **Figure 3b** displays typical right circularly-polarized (σ) PL spectra of MAPbBr₃ SC upon left (σ_+) and right (σ_-) circularly-polarized excitation with 532 nm laser at 77 K. It can be clearly observed that the PL helicity follows that of the optical pumping, a signature of the optically pumped valley polarization. The degree of circular polarization of the PL is defined as:^{48,49}

$$P = \frac{|I(\sigma_+) - I(\sigma_-)|}{I(\sigma_+) + I(\sigma_-)}, \quad (1)$$

where $I(\sigma_+)$, $I(\sigma_-)$ are the PL intensity with left- and right-circular optical pumping, respectively. *P*

was found to be high near Peak 1 ($\sim 8\%$), which however drops to only $\sim 3\%$ at Peak 2. The decrease of P is a consequence of the dominance of the direct transitions at Peak 1 and the indirect transitions at Peak 2. The latter has a much slower rate compared to that of the spin-flipping, smearing out all the polarization information. P also drops significantly when the excitation wavelength is changed from near-resonance to off-resonance. For example, P decreases to 3% and 0% with 473 nm and 400 nm optical-pumping, respectively. This is attributed to the potential barrier formed between the two spin-split bands that preserves the initial photo-carrier spin with near-resonance excitation. Similar results were obtained for CsPbBr₃ SC (**Figure S7**). Recently, Niesner *et al* also observed that the photocurrent of MAPbI₃ SC is highly dependent on the light helicity when excited near the indirect bandgap, namely, the circular photogalvanic effect.⁵⁰ Due to the strong SOC, the spin lifetime in lead halide perovskites is very fast (\sim ps).⁵¹ The observation of optical-pump polarization dependent PL unambiguously point to the spin-split bands near the band edge that extend the carrier spin lifetimes.

Figure 3. PL Helicity upon circularly-polarized excitation. (a) Schematic of the Rashba split bands and selective excitation with circularly-polarized light. (b) Right circularly-polarized (σ_-) PL spectra of MAPbBr₃ SC upon left (σ_+) and right (σ_-) circularly-polarized excitation with 532 nm laser at 77 K. (c) The degree of circular polarization (top panel) detected and (bottom panel) excited at different energies.

Reviewer #5 (Remarks to the Author):

“While this manuscript contains some interesting findings, it cannot be published as the main conclusions are (partially) based on a misinterpretation of spectroscopic measurements.

The authors report a difference in band gap for MAPbBr single crystals and thin films. However, a recent paper in Nature Communications showed very clear evidence that the optical band gap for MAPbBr in single crystals is identical to the band gap in thin films:

<https://www.nature.com/articles/s41467-017-00567-8>

This is in direct contradiction to the results in Fig. 1d. In the manuscript at hand, the problem arises most probably from using diffuse reflectance measurements to determine the band gap of the single crystals. Diffuse reflectance measurements depend on parameters such as grain diameter, surface roughness, and refractive index. Deriving the band gap of a material from diffuse reflectance measurements is therefore likely to be inaccurate, see e.g.:

<http://www.sciencedirect.com/science/article/pii/S0927024807001948>

I note that several groups in the perovskite community have used diffuse reflectance spectroscopy for (doped) perovskite crystals and derived too small band gaps. In general, I want to point out the importance of carefully executed and properly interpreted optical spectroscopy. To obtain the optical properties of single crystal samples, it is advisable to use a specular reflection technique (such as ellipsometry). For thin films transmittance is often a good choice. Diffuse reflectance should only be used for very fine powders and relatively weak absorption.

With respect to the PL measurements, I am not convinced that the apparent peak shift is not due to self-absorption. A clear hint is the reported shift with increased precursor concentration (and hence higher film thickness).”

Response: We thank the reviewer for raising the important questions. However, we believe there are possible some misunderstandings that obfuscate his/her evaluation on the manuscript.

Firstly, we did not report a difference in the optical band gaps of perovskites in thin film or single crystal form. Instead, we show that there are weak absorption tails below the typically measured band gap. Due to the weak-transition characteristics, the absorption is usually not obvious in thin film, leading to the difference in the measured band gap of single crystal and thin film. Nonetheless, the band gap of perovskite thin film and single crystal should be identical, but there was a lot of misinterpretation previously.

Secondly, we agree with the reviewer that diffuse reflectance measurements can be sensitive to many parameters. In our manuscript, we also pointed out any sub-gap defects (corresponding to grain condition, surface roughness as the reviewer points out) that could also have weak absorption and may overlap with the indirect band gap absorption, leading to inaccurate estimation of the real band gap in lead halide perovskites. Hence, we did not attribute the weak absorption immediately to the indirect band gap of the perovskites. Instead, we carefully performed further complimentary studies to validate our claim of indirect band gap: (a) the appearance of a second PL peak at the claimed

indirect band gap position; (b) the second PL peak have a 2nd order fluence dependence, implying band edge recombination; and (c) a strong photo-current peak at similar energy point. These results clearly support the weak absorption and emission are from the indirect band gap of lead halide perovskites.

For accurate measurement of the weak absorption coefficients near the indirect band gap, other techniques have been employed in previous reports. For example, Wang *et al* have successfully fitted the weak absorption part of the photo-thermal diffraction spectroscopy (PDS) data of MAPbI₃ film with an indirect band gap (*Energy Environ. Sci.*, **2017**, 10, 509-515). The diffuse reflectance data of MAPbI₃ SC measured in our previous paper (*Adv. Energy Mater.* **2016**, 1600551) is in good agreement with the PDS data. The only disadvantage of diffuse reflectance spectroscopy is it cannot tell the absorption coefficients of single crystals. However, in this manuscript, we are not quantifying these parameters. Hence, we believe that it is not necessary to further perform more delicate experiments on this part.

Thirdly, the reviewer pointed out: “A clear hint is the reported shift with increased precursor concentration (and hence higher film thickness),” We respectfully disagree with this comment. The thickness of the films has been confirmed for several batches of samples. The thickest films (1M, highest concentration) always have a thickness of around 200 nm; while the other films generally have a thickness between 150 nm – 200 nm. There is not much difference in the thickness of the films with different precursor concentration. And with the known absorption coefficients of MAPbBr₃, it is easy to calculate that there is negligible reabsorption in the thin films. Similar results were obtained from other literature (e.g., *Energy Environ. Sci.*, 2017, 10, 509; *Physical Review B* 2017, 95, 075207). We, therefore, have solid evidence to exclude the reabsorption effect in the films, rather than by empirical intuitions.

To avoid ambiguities, we revised the manuscript accordingly:

In Page 4, Line 2 from the bottom:

“We had previously performed diffuse reflectance spectroscopy (DRS) to measure MAPbI₃ SC absorption edge and the results were consistent with that of the photo-thermal diffraction spectroscopy reported elsewhere.^{18,23} Here DRS is used to measure the absorption properties of the lead bromide perovskites and the absorption edges...”

In Page 5, Line 5:

“Peak 2 is **absent** in TF due to its weak absorbing and emitting nature” is changed to “Peak 2 is **not prominent** in TF due to its weak absorbing and emitting nature”

In addition to the response to the reviewers, we made a few small revisions:

In Page 4, Line 6:

“At low temperatures, non-spherical A-site cations and surface/**defects** induced lattice distortion lead to a static centro-symmetry breaking that mainly contributes to the Rashba effect.”

In Page 11, Line 10:

“The temperature-independent term should arise from static centrosymmetry breaking by the non-spherical A-site organic cations (MA, FA), surface distortion or **internal interface distortions** (such as twinning of orthorhombic phase, inclusion of different phases)...”

Reviewers' comments:

Reviewer #5 (Remarks to the Author):

I have indeed misinterpreted the intention of Fig. 1d and my previous report was oversimplifying, my apologies to the authors.

However, I still have problems with the interpretation of their optical spectra.

The authors state in the manuscript that "Peak 2 is not prominent in TF due to its weak absorbing and emitting nature." However, it does appear in PL when the film quality is sufficiently high (Fig. 2a). Following the authors' interpretation, peak 2 should also be prominent in the absorption of this higher quality film. Why is this simple measurement not presented?

I would also like to note that peak 2 was not observed in these high quality MAPbBr₃ films with high PLQEs:

<https://pubs.acs.org/doi/10.1021/acseenergylett.8b00509>

Furthermore, there is some inconsistency regarding film thicknesses. In the response to my previous report, the authors write:

"The thickest films (1M, highest concentration) always have a thickness of around 200 nm; while the other films generally have a thickness between 150 nm – 200 nm."

From the caption of Fig. 2:

"Temperature-dependent PL peak positions for the film prepared with 1 M precursor concentration. This concentration matches that used for single crystal growth. The film has a thickness of around 220 nm, which is thicker than the film presented in Figure 1e (~ 100 nm)."

Reviewer #6 (Remarks to the Author):

The manuscript presents a comprehensive data set on lead-bromide-perovskite single crystals and thin films. While most of the data have been reported before in many previous publications by the authors and other groups, this work puts the information in context and resolves some apparent discrepancies in the literature. Therefore, I recommend publication in Nature Communications in agreement with some of the previous reviewers.

The authors added new experimental data using circularly-polarized photoluminescence spectroscopy. They observe a clear intensity difference between right and left circular-polarized excitation. To exclude experimental artifacts they should show that the effect reverses when the polarization in the detection path is switched from right to left. The circular dichroism vanishes for higher excitation photon energies and the observations are consistent with a Rashba-type splitting at the band edges. Some references should be added which show that circularly-polarized photoluminescence spectroscopy may be used to identify a spin-polarized Rashba-type band structure.

Also in the context of the reabsorption effect some references to two-photon absorption would be helpful, e.g. <https://doi.org/10.1103/PhysRevApplied.7.014001> and <https://doi.org/10.1103/PhysRevB.95.075207>

Reviewer #5

Comment 1. I have indeed misinterpreted the intention of Fig. 1d and my previous report was oversimplifying, my apologies to the authors.

However, I still have problems with the interpretation of their optical spectra.

The authors state in the manuscript that "Peak 2 is not prominent in TF due to its weak absorbing and emitting nature." However, it does appear in PL when the film quality is sufficiently high (Fig. 2a). Following the authors' interpretation, peak 2 should also be prominent in the absorption of this higher quality film. Why is this simple measurement not presented?

Response 1: We thank the reviewer for his frank, open views and are very grateful for his sincere feedback to help us strengthen our manuscript. About the comment on the interpretation of the optical spectra, the indirect absorption is always not comparable to that of direct absorption, hence, we can not see any prominent difference in absorption for different films. However, absorption and emission are two different processes. For emission (photoluminescence), the photoexcited carriers tend to relax to the lower energy states (here $E_{in} < E_d$), therefore, the indirect band edge is much higher populated than the direct band edge. The indirect PL intensity can be comparable to that of direct PL. A good example is germanium (Ge) which shows similar optical properties. Secondly, the photoexcited carriers can interact with the lattice, forming polaron or self-trapped exciton, which may also change the transition probability.

Comment 2: I would also like to note that peak 2 was not observed in these high quality MAPbBr₃ films with high PLQEs: <https://pubs.acs.org/doi/10.1021/acsenergylett.8b00509>

Response: We thank the reviewer for the interesting question. After reading through the paper, we found the PL spectrum is from the control sample (MAPbBr₃ film). The PLQE is ~1%, which is not high as the reviewer had mentioned. Moreover, the effective carrier lifetime of the film used for PL is still not comparable to our best quality thin film (1M, ~ 40 ns). In the paper, as the authors claimed, the control sample shows initial fast decay within tens of ns, followed by a slow component of 47 ns. The effective lifetime (effective lifetime is defined as the time of the PL reaching 1/e of its initial intensity) shall then be less than 47 ns. After we digitize the data from the paper (Figure R1a), we found the effective lifetime of the sample used for PL is only ~20 ns. This data/value does not account for the presence of even faster decays. This is because of the lower system resolution of the TCSPC system (than a Streak Camera system) and we would also like to highlight that we do not see the rise time of the PL data which was somehow left out by the authors. So it is possible that the real effective PL lifetime is much less than 20 ns.

Instead, our streak camera system has a much higher time resolution and the lifetime measured is more accurate. The high quality films in our experiment has a lifetime of around 40 ns. We would also like to point out that the film in our manuscript prepared with 0.5 M precursor concentration has an effective lifetime of ~18 ns, which is more like the control film used in the *ACS Energy Lett* paper. As we see from our results (Figure 2a), the second peak has become much weaker for the film prepared with 0.5 M precursor concentration compared to that of 1 M, which is almost negligible for film with 0.25 M precursor concentration. It is possible that the quality of the films in the ACS

Energy Lett paper is not as high as it is believed to be. Hence, the absence of the dual peak in their data.

Secondly, even the PL peak shown in the *ACS Energy Lett* paper does not show very symmetric single peak (the fitting of the digitized data using single Voigt function is presented in Figure R1b).

Last but not the least, the carrier lifetime and PLQE are relatively good metrics to evaluate whether a film is of high quality or not. But there are too many factors that may affect carrier lifetime and PLQE (e.g., grain size). Hence, there are no perfect methods to compare film quality prepared in different laboratories. Also, as we mentioned in our manuscript, the dual peak emission is sensitive to film preparing conditions, stoichiometric ratio, local environment, etc.

Based on above points, we believe that our conclusion is still valid.

Figure R1. (a) The effective lifetime of the MAPbBr₃ film in the *ACS Energy Lett* paper (PL decay is digitized from Figure 3b in the paper). (b) The PL spectrum digitized from Figure 3a in that paper and its fitting with a single Voigt function.

Comment 3: Furthermore, there is some inconsistency regarding film thicknesses. In the response to my previous report, the authors write: "The thickest films (1M, highest concentration) always have a thickness of around 200 nm; while the other films generally have a thickness between 150 nm – 200 nm."

From the caption of Fig. 2: "Temperature-dependent PL peak positions for the film prepared with 1 M precursor concentration. This concentration matches that used for single crystal growth. The film has a thickness of around 220 nm, which is thicker than the film presented in Figure 1e (~ 100 nm)."

Response 3: We thank the reviewer for pointing out a potential ambiguity. The film prepared with 0.25 M concentration has a thickness of around 150 nm and that prepared with 1M concentration is around 220 nm (Figure R2). The PL in Figure 1e is measured with film prepared using 0.25 M concentration. We apologize for the confusion caused in the caption of Figure 2.

We have changed the caption of Fig.2 to avoid any potential ambiguity: "The film has a thickness of around 220 nm, ~~which is thicker than the film presented in Figure 1e (~100 nm).~~"

In addition, a sentence was added in Figure 1e: "The TF was prepared using 0.25 M precursor

concentration as discussed in detail in the next section.”

Figure R2. Typical lateral SEM image of the films prepared using (a) 0.25 M and (b) 1 M precursor concentration. The film thicknesses are around 150 nm and 220 nm for (a) and (b), respectively.

Reviewer #6

The manuscript presents a comprehensive data set on lead-bromide-perovskite single crystals and thin films. While most of the data have been reported before in many previous publications by the authors and other groups, this work puts the information in context and resolves some apparent discrepancies in the literature. Therefore, I recommend publication in Nature Communications in agreement with some of the previous reviewers.

The authors added new experimental data using circularly-polarized photoluminescence spectroscopy. They observe a clear intensity difference between right and left circular-polarized excitation. To exclude experimental artifacts they should show that the effect reverses when the polarization in the detection path is switched from right to left. The circular dichroism vanishes for higher excitation photon energies and the observations are consistent with a Rashba-type splitting at the band edges. Some references should be added which show that circularly-polarized photoluminescence spectroscopy may be used to identify a spin-polarized Rashba-type band structure.

Also in the context of the reabsorption effect some references to two-photon absorption would be helpful, e.g. <https://doi.org/10.1103/PhysRevApplied.7.014001> and <https://doi.org/10.1103/PhysRevB.95.075207>

Response: We thank the reviewer for his favorable comments on the new version of the manuscript.

For the change of the detection polarization, we had also performed the measurement previously, but had not shown the data. Indeed, the effect reverses as shown in Figure R3.

Figure R3. A comparison of MAPbBr₃ SC PL detected at (a) left σ_+ and (b) right σ_- circular polarizations. The excitation wavelength is 473 nm.

We have incorporated Figure R3 in the revised SI (Figure S7) and made changes in the main manuscript in Page 9, 2nd line from the bottom:

“...optically pumped valley polarization. The same conclusion can be obtained when the detection polarization is reversed (Figure S7). ...”

For references showing circularly-polarized PL spectroscopy can be used to identify spin-polarized band structure, we have added a few references in the manuscript in Page 9, Paragraph 2:

“The technique is widely used to selectively excite spin-valleys to achieve valley polarization, which provides a good indication of how well the valley identity of charge carriers is preserved before recombination.^{36,50,51}” where reference 50, 51, 52 are:

50. Zeng, H. L., Dai, J. F., Yao, W., Xiao, D. & Cui, X. D. Valley polarization in MoS₂ monolayers by optical pumping. *Nat. Nanotechnol.* **7**, 490-493 (2012).
51. Mak, K. F., He, K., Shan, J. & Heinz, T. F. Control of valley polarization in monolayer MoS₂ by optical helicity. *Nat. Nanotechnol.* **7**, 494-498 (2012).
52. Cao, T. *et al.* Valley-selective circular dichroism of monolayer molybdenum disulphide. *Nat. Commun.* **3**, 887 (2012).

We also thank the reviewer for providing some more references which we had missed in our manuscript about reabsorption. We have incorporated them into the manuscript in Page 6, last line:

“exciton-electron scattering (H emission) or biexciton emission; and (7) reabsorption effect.^{20,34-36,} where reference 35, 36 are:

- 35 Yamada, T., Yamada, Y., Nakaike, Y., Wakamiya, A. & Kanemitsu, Y. Photon Emission and Reabsorption Processes in CH₃NH₃PbBr₃ Single Crystals Revealed by Time-Resolved Two-Photon-Excitation Photoluminescence Microscopy. *Phys. Rev. Appl.* **7**, 014001 (2017).
- 36 Niesner, D. *et al.* Temperature-dependent optical spectra of single-crystal CH₃NH₃PbBr₃

cleaved in ultrahigh vacuum. *Phys. Rev. B* **95**, 075207 (2017).

REVIEWERS' COMMENTS:

Reviewer #5 (Remarks to the Author):

The authors have addressed all my comments in great detail. I have no further objections, the manuscript can be published as is.

Reviewer #6 (Remarks to the Author):

The authors show convincingly that the effect reverses for the opposite polarization. The added references on circularly-polarized photoluminescence spectroscopy concern MoS₂ where the two valleys are not related to a Rashba-type spin splitting. I would recommend citing this work instead <https://doi.org/10.1021/acseenergylett.8b00638> which shows "Circularly polarized photoluminescence [...] in organic–inorganic hybrid perovskite films". When preparing my previous report, I did not have this reference at hand.

All the points raised in this round of review have been satisfactorily addressed and the manuscript may be published in Nature Communications after replacing the references as mentioned above.

Reviewer #6 (Remarks to the Author):

The authors show convincingly that the effect reverses for the opposite polarization.

The added references on circularly-polarized photoluminescence spectroscopy concern MoS₂ where the two valleys are not related to a Rashba-type spin splitting. I would recommend citing this work instead <https://doi.org/10.1021/acsenergylett.8b00638>

which shows "Circularly polarized photoluminescence [...] in organic–inorganic hybrid perovskite films". When preparing my previous report, I did not have this reference at hand.

All the points raised in this round of review have been satisfactorily addressed and the manuscript may be published in Nature Communications after replacing the references as mentioned above.

Response: We thank the reviewer for his suggestion. We have inserted the reference in Page 9, Paragraph 2, Line 2:

"...The technique is widely used to selectively excite spin-valleys to achieve valley polarization, which provides a good indication of how well the valley identity of charge carriers is preserved before recombination.⁵⁰⁻⁵³ The experimental...",

where reference 53 is:

53. Myung, C. W., Javid, S., Kim, K. S. & Lee, G. Rashba–Dresselhaus Effect in Inorganic/Organic Lead Iodide Perovskite Interfaces. *ACS Energy Lett.* **3**, 1294-1300 (2018).

We thank Reviewer 4 for alerting us to the ArXiv paper. We also found out that this ArXiv paper was recently published online in the Journal of Physical Chemistry C (DOI: 10.1021/acs.jpcc.8b11288). Based on the suggestion by Reviewer 4, we added a discussion on the recently posted work on the JPCC paper., as well as another recently published paper in Phys. Rev. B, both showing theoretical evidence of dynamical Rashba splitting effect in CsPbI₃. On Page 13, last paragraph, we added:

Amidst the review period of this manuscript, two recent theoretical reports on the calculated dynamical Rashba splitting energy for CsPbI₃ came to our attention.⁶¹⁻⁶² At room temperature, the calculated dynamical Rashba split is around 10 to 20 meV, which is approximately one order smaller than our experimental results. We attribute this difference to the more significant atomic vibrations of the lighter bromine compared to iodine. Furthermore, imperfect lattice, polaron formation, phonons involved in the indirect emission processes and the approximations in the calculation methods may also account for the difference. Nonetheless, these theoretical reports lend crucial support for the dynamical Rashba effect.

where references 61 and 62 are:

61. Marronnier, A. *et al.* Influence of Disorder and Anharmonic Fluctuations on the Dynamical Rashba Effect in Purely Inorganic Lead-Halide Perovskites. Published Online DOI: 10.1021/acs.jpcc.8b11288 (2018)

62. Mckechnie, S. *et al.* Dynamic symmetry breaking and spin splitting in metal halide perovskites. *Phys. Rev. B* **98**, 085108 (2018).